# *LINC00842* inactivates transcription co-regulator PGC-1α to promote pancreatic cancer malignancy through metabolic remodelling

Xudong Huang[1,7], Ling Pan[1,7], Zhixiang Zuo [1,7], Mei Li[2,7], Lingxing Zeng[1], Rui Li[1], Ying Ye[1], Jialiang Zhang[1], Guandi Wu[1], Ruihong Bai[1], Lisha Zhuang[1], Lusheng Wei[3], Yanfen Zheng[1], Jiachun Su[1], Junge Deng[1], Shuang Deng[1], Shaoping Zhang[1], Shihao Zhu[1], Xu Che[4], Chengfeng Wang[4], Chen Wu [5], Rufu Chen[3,6], Dongxin Lin[1,5 ✉] & Jian Zheng [1✉]

The molecular mechanism underlying pancreatic ductal adenocarcinoma (PDAC) malignancy remains unclear. Here, we characterize a long intergenic non-coding RNA *LINC00842* that plays a role in PDAC progression. *LINC00842* expression is upregulated in PDAC and induced by high concentration of glucose via transcription factor YY1. *LINC00842* binds to and prevents acetylated PGC-1α from deacetylation by deacetylase SIRT1 to form PGC-1α, an important transcription co-factor in regulating cellular metabolism. *LINC00842* over-expression causes metabolic switch from mitochondrial oxidative catabolic process to fatty acid synthesis, enhancing the malignant phenotypes of PDAC cells. High *LINC00842* levels are correlated with elevated acetylated- PGC-1α levels in PDAC and poor patient survival. Decreasing *LINC00842* level and inhibiting fatty acid synthase activity significantly repress PDAC growth and invasiveness in mouse pancreatic xenograft or patient-derived xenograft models. These results demonstrate that *LINC00842* plays a role in promoting PDAC malignancy and thus might serve as a druggable target.

[1] Sun Yat-sen University Cancer Center, State Key Laboratory of Oncology in South China and Collaborative Innovation Center for Cancer Medicine, Guangzhou, China. [2] Department of Pathology, Sun Yat-sen University Cancer Center, Guangzhou, China. [3] Department of Pancreaticobiliary Surgery, Sun Yat-sen Memorial Hospital, Sun Yat-Sen University, Guangzhou, China. [4] Department of Abdominal Surgery, National Cancer Center/National Clinical Research Center/Cancer Hospital, Chinese Academy of Medical Sciences and Peking Union Medical College, Beijing, China. [5] Department of Etiology and Carcinogenesis, National Cancer Center/National Clinical Research Center/Cancer Hospital, Chinese Academy of Medical Sciences and Peking Union Medical College, Beijing, China. [6] Guangdong Provincial People's Hospital & Guangdong Academy of Medical Sciences, Guangzhou, China. [7] These authors contributed equally: Xudong Huang, Ling Pan, Zhixiang Zuo, Mei Li. ✉email: lindx@sysucc.org.cn; zhengjian@sysucc.org.cn

Pancreatic ductal adenocarcinoma (PDAC) is one of the most fatal malignancies with 5-year survival rates <9%[1,2]. Metastasis often already occurs at PDAC diagnosis and is responsible for the high mortality rate of the disease[3]. A larger amount of experimental evidence has demonstrated that metastasis is a multistage process involving many genes and pathways and the tumor microenvironment[4], but the specific and detailed mechanism for PDAC metastasis is poorly understood yet. Discovering the molecular mechanisms underlying pancreatic cancer cell invasion and metastasis is essential for developing effective therapy and thus improving patients' survival.

Recent studies have shown that many oncogenes and mutated tumor suppressor genes, such as c-MYC and P53, are involved in modulating tumor cell metabolism[5,6]. In addition, altered by gene mutation or expressional aberration, some metabolic enzymes such as succinate dehydrogenase, fumarate hydratase, pyruvate kinase, and isocitrate dehydrogenase have also been linked to cancer development and progression[7]. Therefore, cancer cells tend to reprogram their metabolism and energy production pathways to provide themselves with necessary bioenergetics and biosynthetic intermediates for rapid proliferation and invasiveness[8,9]. These findings open an avenue to cancer treatment by targeting the specific metabolic enzymes or metabolic pathways.

Long intergenic non-coding RNAs (lincRNAs), a class of biological process regulators, have been shown to play a role in metabolic regulation[10–12]. For example, lincRNAs may recruit transcriptional factors to regulate the expression of genes involved in metabolic processes[13]; may regulate posttranslational modification of metabolism-related proteins or may function as scaffolds or decoys to promote interactions between metabolic enzymes[14–17]. It has been documented that many lincRNAs are aberrantly expressed in human cancer including PDAC (refer to The Cancer Genome Atlas database). However, the effects of aberrantly expressed lincRNAs on metabolism remodeling of PDAC have rarely been investigated, especially whether they play a role in PDAC invasion and metastasis.

Peroxisome proliferator-activated receptor gamma coactivator 1-alpha (PPARGC1A or PGC-1α) is a master transcriptional co-regulator participating in remodeling the metabolic landscape[18]. Aberrant expression and posttranslational modifications of PGC-1α have strong impact on its activity as the transcriptional regulator[18,19]. Previous work reported that the posttranslational acetylation modification of PGC-1α is catalyzed by acetyltransferase GCN5[20] and deacetylase SIRT1[21]. Acetylated PGC-1α is the inactive form of PGC-1α and deacetylation will make this protein to recruit other transcriptional factors or coregulators and then initiate transcription of its target genes[19]. Although dysregulation of PGC-1α/MYC expression has been shown to play a role in maintaining the stemness property of pancreatic cancer stem cells[22], the regulation and function of PGC-1α acetylation modification in pancreatic cancer are poorly understood.

In this work, we examine the roles of PDAC-related lincRNAs, which are identified by mining The Cancer Genome Atlas (TCGA) PDAC RNA-sequencing data, in metabolic remodeling and consequent PDAC malignancy. We find that the expression levels of LINC00842 are significantly higher in PDAC tumors compared with normal tissues and high LINC00842 levels are significantly correlated with poor survival in patients. We also demonstrate that LINC00842 causes metabolic remodeling in PDAC cells through interacting with the transcription co-factor PGC-1α, which prevents acetylated PGC-1α protein from deacetylation by SIRT1. Ectopic LINC00842 overexpression promotes proliferation and invasiveness of PDAC in vitro and in vivo in mice. Furthermore, we perform experimental treatment in mouse patient-derived xenograft (PDX) models and the results suggest that LINC00842 may be a potential therapeutic target for PDAC.

## Results

**Identification of PDAC-related LINC00842.** We started searching the lncRNAs potentially associated with PDAC prognosis in TCGA PDAC RNA-sequencing data and found that among the 1908 lncRNAs expressed in PDAC samples, 42 were significantly associated with survival time in patients (FDR $P \leq$ 0.05; Fig. 1a and Supplementary Data 1). We selected 10 top significant lncRNAs to verify their associations with survival time in our PDAC patients recruited at Sun Yat-sen Memorial Hospital and Sun Yat-sen University Cancer Center (Cohort 1, see the "Methods" section) and found that only the levels of LINC00842, which is modestly conserved across different species (Supplementary Fig. 1a), were significantly associated (Fig. 1b). Kaplan–Meier estimation and multivariate Cox proportional hazard model analysis in Cohort 1 and Cohort 2, recruited at Cancer Hospital Chinese Academy of Medical Sciences (see "Methods"), showed that patients with high LINC00842 level (≥median) had shorter survival time than patients with low LINC00842 level (<median) (Fig. 1c). A similar result was also obtained by analyzing the International Cancer Genome Consortium (ICGC) PDAC dataset (Supplementary Fig. 1b). Quantification of LINC00842 in clinical tissue samples revealed that the levels were significantly higher in tumors than in adjacent normal tissues (Fig. 1d) and in stages III/IV than in stages I/II PDAC (Fig. 1e). We verified the presence of a transcript size as predicted by NCBI annotation (NR_033957.2) in PDAC cell lines by Northern blotting (Supplementary Fig. 1c) and found that PANC-1 and SW1990 cells contained 193 ± 10 ($n = 3$) and 143 ± 5 ($n = 3$) copies of LINC00842 per cell, respectively (Supplementary Fig. 1d). In silico analysis of ribosome nascent chain complex (RNC)-sequencing and ribosome-sequencing data[23] revealed that similar to classic lncRNAs XIST and NEAT1, LINC00842 has low affinity for ribosome (Supplementary Fig. 1e), suggesting that LINC00842 may not have protein-producing ability and is a bona fide lncRNA involved in PDAC prognosis.

We then explored the role of LINC00842 by disturbing its expression in PDAC cell lines. We used CRISPR/Cas9 system to knock in a 3 × poly (A) transcription stop cassette to the 5′ region of LINC00842 locus to silence its transcription and used a lentivirus system to ectopically overexpress LINC00842 (Supplementary Fig. 1f–h and Supplementary Table 1). Overexpressing LINC00842 significantly increased the abilities of PDAC cell proliferation, colony formation, migration, and invasion in vitro while silencing LINC00842 had opposite effects (Fig. 1f–i, Supplementary Fig. 2a and 2b). Assays in mice also showed that subcutaneous xenografts derived from PDAC cells overexpressing LINC00842 had significantly increased growth rates while xenografts derived from PDAC cells with LINC00842 silence had significantly reduced tumor growth rates compared with each control (Fig. 1j). We also implanted PDAC cells into mice pancreas and tested the effect of LINC00842 expression change on tumor metastasis. The results showed that LINC00842 overexpression significantly increased distant metastasis of PDAC cells and reduced animal survival time while LINC00842 silence had opposite effects (Fig. 1k–o and Supplementary Fig. 2c). We also found that depletion of LINC00842 RNA in PANC-1 and SW1990 cells by using antisense oligonucleotide (ASO) of LINC00842(Supplementary Fig. 3a) also significantly suppressed cell proliferation, migration, and invasion compared with control (Supplementary Fig. 3b and 3c). These results clearly demonstrated that LINC00842 acts as an oncogenic lncRNA that promotes PDAC progression and invasiveness.

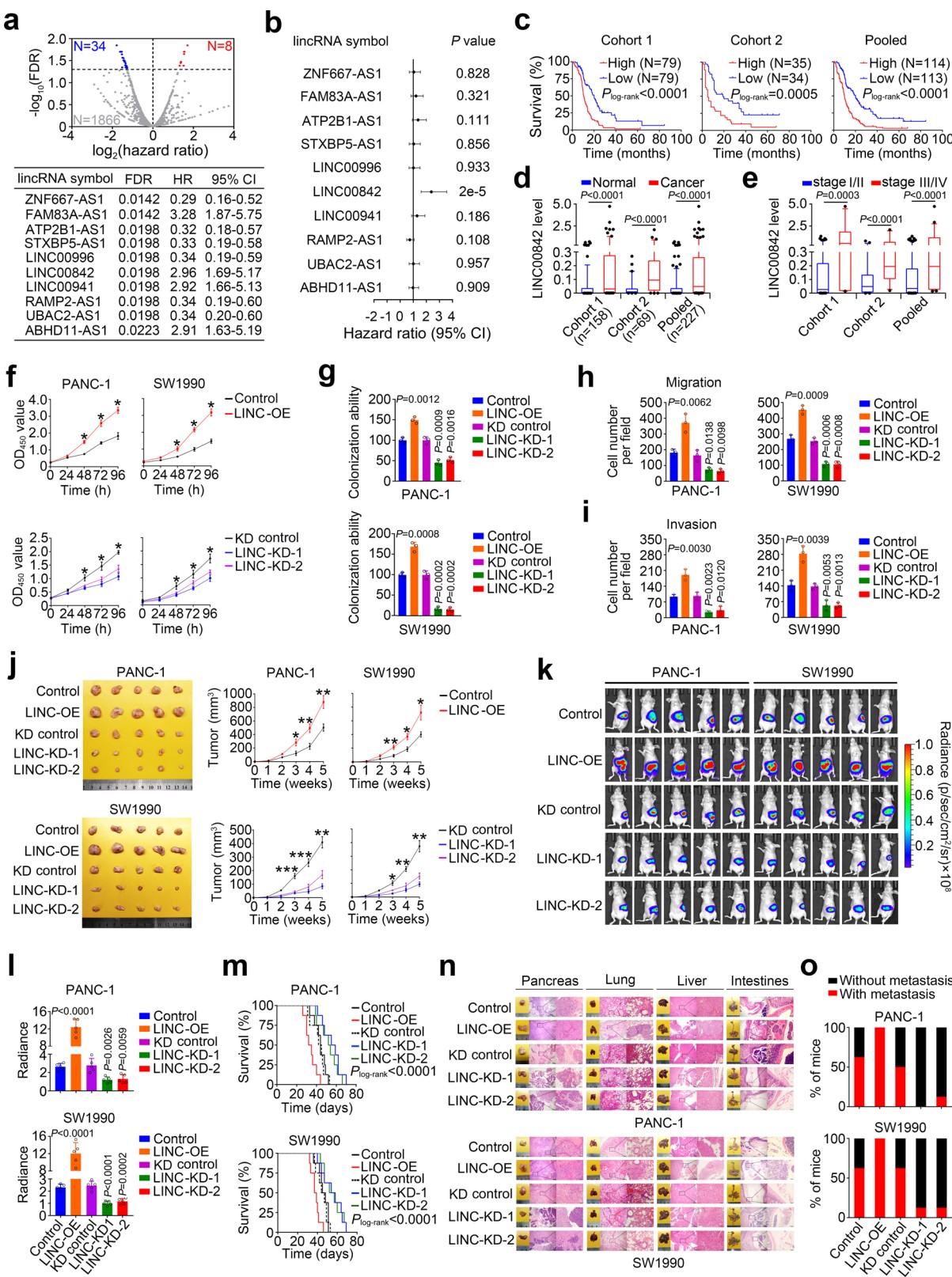

**LINC00842 overexpression leads to metabolic remodeling in PDAC cell.** We sought to elucidate the action mechanism in which *LINC00842* provokes PDAC cell growth and invasiveness. First, we looked at transcriptomic alterations in PANC-1 and SW1990 cells with *LINC00842* silence and identified 270 genes whose expressions were upregulated (fold change ≥2) and 90 genes whose expressions were downregulated (fold change ≤0.5)

(Fig. 2a). Most of these genes were the components of the mitochondrial catabolic programs, such as citrate cycle, pyruvate metabolism, oxidative phosphorylation and fatty acid oxidation, or the anabolic process pathways, including glycolysis/gluconeogenesis and fatty acid biosynthesis (Fig. 2b). The expression alterations of some of these genes, such as tricarboxylate transport protein (*SLC25A1*) and fatty acid synthase (*FASN*) due to

**Fig. 1 *LINC00842* is associated with survival time in patients with PDAC. a** Volcano plot of lncRNAs associated with survival time in TCGA PAAD patients. Red and blue dots represent FDR ≤ 0.05 while gray dots represent FDR > 0.05 (upper panel). Ten top significant lncRNAs selected for investigation (lower panel). **b** Associations of the expression levels of 10 lncRNAs with survival time of patients in Cohort 1, showing only *LINC00842* was significantly associated. HR, hazard ratio; CI, confidence interval. **c** Kaplan–Meier estimates of survival time of patients in Cohort 1, Cohort 2 and combined sample by different *LINC00842* levels in tumor. The median survival times for patients with high *LINC00842* (≥ median) in Cohort 1, Cohort 2 and combined sample were 10.3, 5.0, and 9.9 months, significantly shorter than 20.0, 19.0, and 20.0 months in patients with low *LINC00842* (< median), with the HRs (95% CI) for death of high *LINC00842* level being 2.39 (1.60–3.56), 2.58 (1.13–5.90), and 2.45 (1.77–3.40), respectively. **d**, **e** *LINC00842* levels were significantly higher in PDAC tumors than paired normal tissues (**d**, Cohort 1 ($n = 158$), Cohort 2 ($n = 69$) and combined sample ($n = 227$)) and in stage III/IV tumors (Cohort 1 ($n = 22$), Cohort 2 ($n = 22$) and combined sample ($n = 44$)) than in stage I/II tumors (Cohort 1 ($n = 136$), Cohort 2 ($n = 47$) and combined sample ($n = 183$)) (**e**). Data are shown in box plots; the lines in the middle of the box are medians and the upper and lower lines indicate 25th and 75th percentiles. Wilcoxon rank-sum tests (two-tailed) were used in (**d**) and (**e**) to determine the *P* values. The number of stage I/II tumors are 136 and 47 and stage III/IV tumors are 22 and 22 in Cohort 1 and Cohort 2, respectively. **f–i** Effects of *LINC00842* overexpression (LINC-OE) or silence (LINC-KD) on abilities of PDAC cell proliferation (**f**, results are means ± SD from 3 experiments and each had 6 replicates), colony formation (**g**, results are means ± SD from 3 independent experiments), migration (**h**) and invasion (**i**). Data in (**h**) and (**i**) are mean ± SD from 3 random fields. **j** Effect of *LINC00842* expression change on PDAC xenograft growth in mice. Shown are images of the subcutaneous xenografts at the end of experiments (left panels) and the growth curves of the xenografts (right panels). Data represent mean ± SEM ($n = 5$). **k** Bioluminescence images showing the effect of *LINC00842* expression change on the tumor burden of mice with orthotopically transplanted PDAC ($n = 5$). **l** Statistics of fluorescence intensity of tumor burden in mice shown in (**k**). Data represent mean ± SD. **m** Effect of *LINC00842* expression change on survival time of mice with PDAC orthotopic transplantation (n = 8). **n`** Histopathological images of 4 organs of mice with orthotopic transplantation show differences in tumor metastases (H&E staining). Scale bars, 500 and 100 μm. **o** Statistics of tumor metastases in mice in each group. The *P* values in (**g–i**), (**l**) and *P < 0.05; **P < 0.01, and ***P < 0.001 in (**f**), (**j**) were determined by Student's *t*-test (two-tailed) compared with corresponding control.

*LINC00842* expression change, were independently verified by using quantitative RT-PCR (Fig. 2c). These results suggested participation of *LINC00842* in metabolic remodeling. We then measured mitochondrial function and aerobic glycolysis in PDAC cells to test the effects of *LINC00842* change on glucose metabolic processes and found that the oxygen consumption rates (OCRs) were significantly decreased in cells overexpressing *LINC00842* but significantly increased in cells with *LINC00842* expression silenced (Fig. 2d). Cells overexpressing *LINC00842* had significantly increased extracellular acidification rate (ECAR), glucose uptake, and lactate production while cells with *LINC00842* silence had significantly decreased levels of these markers (Fig. 2e and f). We further examined lipid contents in these cells and found that *LINC00842* overexpression significantly increased cellular lipids while *LINC00842* silence significantly decreased cellular lipids compared with control cells (Fig. 2g and h). The similar results were obtained in cells where *LINC00842* was depleted by ASOs (Supplementary Fig. 3d–g). Finally, we profiled the metabolites in PDAC cells and the results showed that the metabolites from the tricarboxylic acid cycle (TCA cycle) and long-chain acylcarnitines were significantly decreased in cells overexpressing *LINC00842* while significantly increased in cells with *LINC00842* silenced (Fig. 3a–d). Specifically, by using $^{13}$C-U$_6$-glucose, we found that M + 2 metabolites were significantly reduced while M + 3 metabolites were significantly increased in cells overexpressing *LINC00842* compared with that in control cells; however, opposite results were detected in cells with *LINC00842* silenced (Fig. 3e–h). These results suggest that in PDAC cells overexpressing *LINC00842*, the rate of TCA cycle through pyruvate dehydrogenase was repressed, leading to decreased glucose oxidation in the mitochondria; however, the metabolic process switched to the anaplerotic reaction via pyruvate carboxylase. For lipid metabolism, we observed significantly increased incorporations of $^{13}$C glucose carbon pairs to palmitate (C16:0) and stearate (C18:0) in cells overexpressing *LINC00842* (Fig. 3i–l) while the incorporations were significantly declined in cells with *LINC00842* silenced, suggesting a role of *LINC00842* in enhancing lipid synthesis from glucose. Furthermore, we examined citrate levels in PDAC cell mitochondria and cytosol and found that *LINC00842* overexpression significantly increased citrate accumulation in the cytosol (source of *de novo* lipid synthesis) but significantly decreased citrate level in the

mitochondria. However, citrate was accumulated in the mitochondria in cells with *LINC00842* depletion (Supplementary Fig. 4). These results may explain why PDAC cells overexpressing *LINC00842* had enhanced lipid synthesis by using accumulated cytosolic citrate despite total citrate level was decreased (Fig. 3a and c). Together, these results indicated that *LINC00842* overexpression alters PDAC cell metabolic program.

**_LINC00842_ targets PGC-1α**. We next wanted to characterize the molecular mechanism underlying the role of *LINC00842* in remodeling the metabolic programs in PDAC cells. Since >70% of *LINC00842* was present in the nucleus (Fig. 4a and b), we performed RNA pull-down assays using *LINC00842* sense probe and nuclear lysates from PANC-1 cell lines; the pull-down products were then analyzed by mass spectrometry to determine any specific *LINC00842*-proteins interactions. We identified 116 potential interacting proteins (Supplementary Table 2) and selected 8 tops ranked by the exponentially modified protein abundance index (emPAI) for Western blot validation. We found that among the 8 proteins, only PGC-1α interacted with *LINC00842* in both PANC-1 and SW1990 cells (Fig. 4c). RNA immunoprecipitation with the antibody against PGC-1α verified specific interaction between *LINC00842* and PGC-1α in cells (Fig. 4d). Chromatin isolation by RNA purification (ChIRP) assays showed that PGC-1α can be precipitated by biotin-labeled *LINC00842* antisense probes (Fig. 4e). Furthermore, fluorescence in situ hybridization showed that *LINC00842* is co-localized with PGC-1α in the nucleus of PDAC cells (Fig. 4f). All these results pointed out that *LINC00842* may interact with PGC-1α in PDAC cells. We then constructed various truncated *LINC00842* and truncated *PPARGC1A* (Fig. 4g and h) to examine the specific domains required for the interaction and the results showed that the 5′-end of *LINC00842* (nucleotides 1–690) and the PGC-1α Repression domain (aa181–460) are required for the interaction of these two molecules (Fig. 4g and i). Moreover, bioinformatics analysis revealed that there are 6 positions within 5′-end of *LINC00842* (nucleotides 1–690) that might require for the interaction (Supplementary Fig. 5a). RNA pull-down assays showed that mutations in these positions substantially abolished the interaction of *LINC00842* with PGC-1α (Supplementary Fig. 5b). RIP assays with antibody against PGC-1α showed that treatment of RNase A but not RNase III significantly reduced the

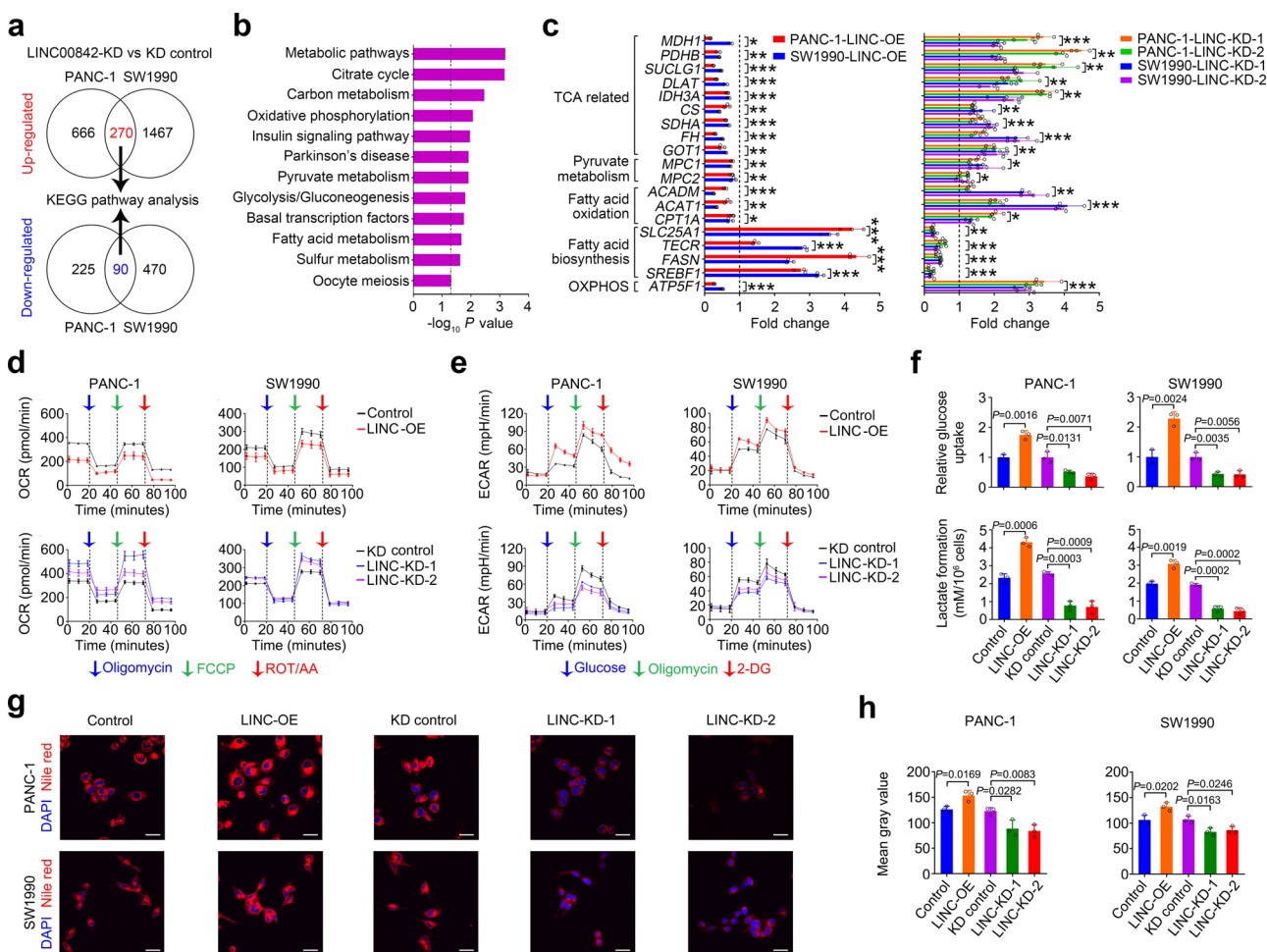

**Fig. 2 LINC00842 alters metabolic programs in PDAC cells. a** Differentially expressed genes identified by RNA sequencing in PDAC cells with or without *LINC00842* knockdown (KD). Upregulated genes (fold change ≥2) and downregulated genes (fold change ≤0.5) were selected for pathway enrichment analysis. **b** Kyoto Encyclopedia of Genes and Genomes (KEGG) analysis of differentially expressed genes regulated by *LINC00842*. The dotted line indicates *P* < 0.05 (two-tailed). **c** Validation of some RNA-sequencing results by using quantitative RT-PCR. Data were normalized to control or KD-control. TCA, tricarboxylic acid cycle; OXPHOS, oxidative phosphorylation. **d, e** Effects of *LINC00842* overexpression (LINC-OE) or silence (LINC-KD) on oxygen consumption rate (OCR) (**d**) or extracellular acidification rate (ECAR) (**e**). **f** Effects of *LINC00842* expression change on glucose uptake or lactate production. **g, h** Representative pictures of Nile red (red) and DAPI (blue) staining of cells with *LINC00842* overexpression or silencing (**g**) and quantification using Image J software (**h**). Scale bar, 30 μm. Results in (**c**), (**f**), and (**h**) are mean ± SD from 3 independent experiments. The *P* value in (**f**), (**h**) and *\*P* < 0.05; \*\**P* < 0.01, and \*\*\**P* < 0.001 in (**c**) were determined by Student's *t*-test (two-tailed).

*LINC00842* enrichment (Supplementary Fig. 6a), indicating that it is single-strand *LINC00842* that interacts with PGC-1α since only single-strand RNA, but not double-strand RNA, is subject to degradation by RNase A. The RNA EMSA assays in vitro showing that *LINC00842* interacts with PGC-1α to form an RNA–protein complex (Supplementary Fig. 6b) further verified the RIP and ChIRP results (Fig. 4d and e).

**LINC00842 binding prevents acetylated PGC-1α from deacetylation by SIRT1.** Since PGC-1α is a transcriptional coactivator that regulates expressions of genes involved in the energy metabolism[18], we therefore explored how *LINC00842* may act on PGC-1α. We found that forced changes of *LINC00842* levels in PANC-1 and SW1990 cells did not significantly alter *PPARGC1A* RNA and protein levels (Fig. 5a), suggesting that the function of *LINC00842* may not through affecting *PPARGC1A* transcription and translation. However, *LINC00842* substantially increased PGC-1α acetylation, a posttranslational modification necessary for the transcriptional

activity of PGC-1α[21] (Fig. 5b, Supplementary Fig. 6c). Further assays showed that although *LINC00842* did not influence the expression levels of PGC-1α acetyltransferase *KAT2A* (also named GCN5)[20] (Supplementary Fig. 6d) and deacetylase SIRT1[21] (Fig. 5c), change of *LINC00842* expression specifically impacted the interaction of PGC-1α with SIRT1 (Fig. 5d) but not with GCN5 (Supplementary Fig. 6e): *LINC00842* over-expression decreased but silence increased the PGC-1α and SIRT1 interaction (Fig. 5d) and the decreased interaction can be restored by treatment with RNase A but not RNase III in cells with *LINC00842* overexpression (Supplementary Fig. 6f). We then treated PDAC cells with SRT2104, a SIRT1 agonist, and found that the level of interaction between *LINC00842* and PGC-1α or PGC-1α Repression domain was substantially declined in a dose-dependent manner (Fig. 5e–g), suggesting that *LINC00842* may specifically interact with acetylated PGC-1α. We further found that although SRT2104 treatment reduced acetylated PGC-1α level, SRT2104 treatment had no significant effect on acetylated PGC-1α levels in cells overexpressing *LINC00842* (Fig. 5h), supporting the notion that *LINC00842*

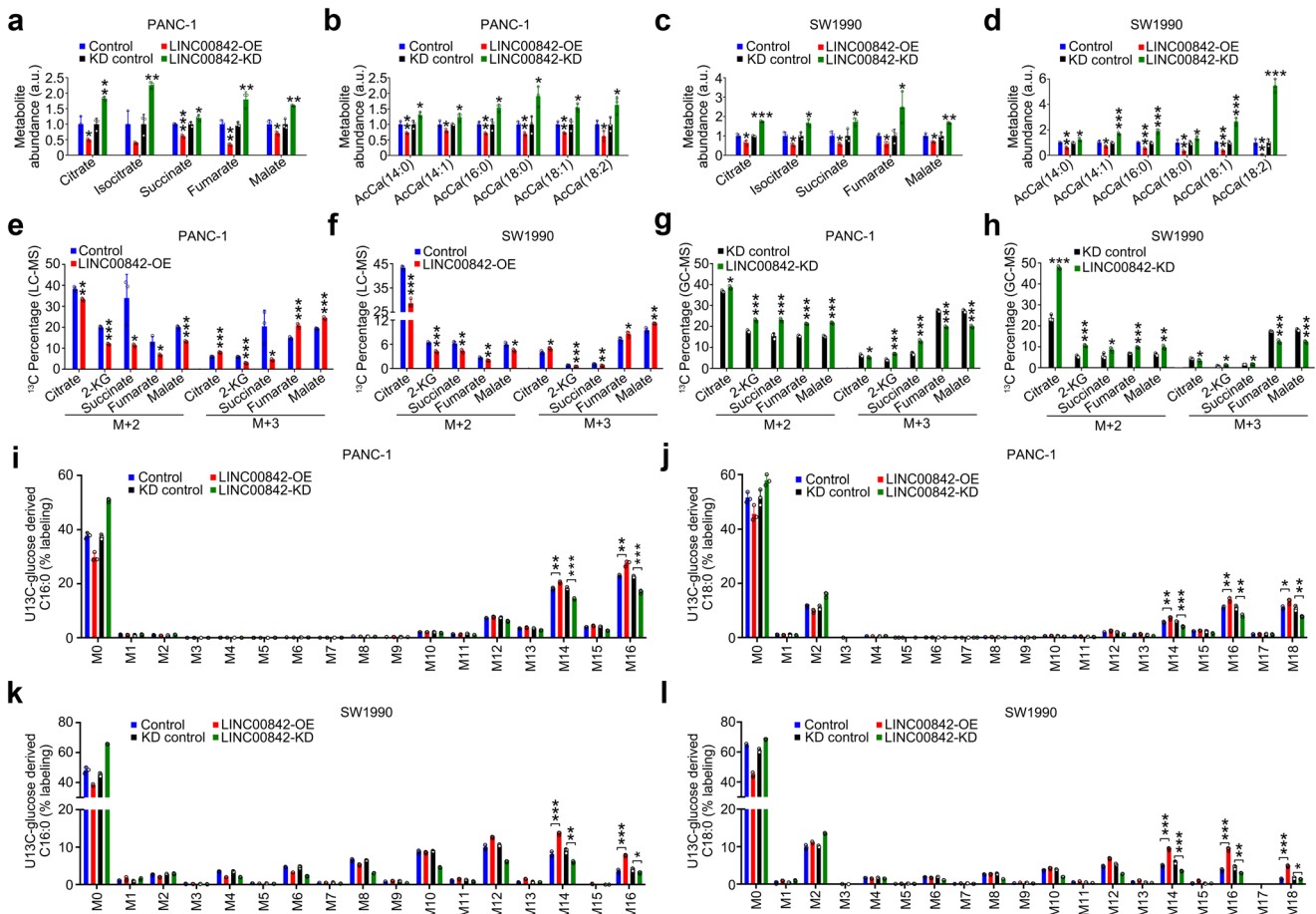

**Fig. 3 LINC00842 promotes anabolic metabolism in PDAC cells. a–d** LINC00842 expression change significantly altered the abundance of tricarboxylic acid cycle (TCA) intermediates (**a** and **c**) or long-chain acylcarnitines (**b** and **d**). **e–h** LINC00842 overexpression (OE) or knockdown (KD) significantly changed the levels of TCA intermediates expressed as mass isotopomer percentage. **i–l** LINC00842 expression change significantly altered the levels of palmitate (C16:0, **i** and **k**) and stearate (C18:0, **j** and **l**) paired mass isotopomer percentages. Results in (**a–l**) are mean ± SD from 3 independent experiments. *$P < 0.05$; **$P < 0.01$, and ***$P < 0.001$ were determined by Student's t-test (two-tailed).

inhibits PGC-1α deacetylation by obstructing access of deacetylase SIRT1 to PGC-1α. An in vitro deacetylation assay showed that LINC00842 did specifically block SIRT1 to deacetylate PGC-1α (Fig. 5i). To further confirm the interaction of LINC00842 with acetylated PGC-1α, we constructed a mutant type of PGC-1α contain 13 acetylation site mutations (lysine to arginine, K to R), a transcriptional activity type which was not affected by acetylation. Immunoprecipitation assay confirmed the mutations substantially abolished PGC-1α acetylation (Supplementary Fig. 6g). RIP and RNA pull-down assays indicated that mutations at the PGC-1α acetylation sites abolished the interaction of the protein with LINC00842 as compared with wild-type PGC-1α (Supplementary Fig. 6h and 6i). In addition, we investigated the association between PPARGC1A mRNA levels in PDACs and survival time in patients in TCGA and ICGC cohorts and the results were negative (Supplementary Fig. 7a and 7b), further suggesting that the functional effect of LINC00842 on PGC-1α is not at the transcriptional level but via posttranslational acetylation modification.

It has been reported that PGC-1α participates in fatty acid synthesis[24–27]. We found that in PDAC cells, PGC-1α overexpression significantly decreased but knockdown significantly increased the expression levels of SLC25A1 and FASN, two genes that play important roles in fatty acid synthesis (Supplementary Fig. 7c and 7d). Nile red staining further indicated that PGC-1α

overexpression significantly decreases but its silence significantly increases cellular lipids compared with control cells (Supplementary Fig. 7e and 7f). We then carried out rescue assays in PDAC cells to determine whether PGC-1α participates in metabolic remodeling caused by LINC00842. In PDAC cells with LINC00842 silence, depletion of PPARGC1A restored cell proliferation, migration, and invasion abilities (Supplementary Fig. 8a and 8b); in parallel, depletion of PPARGC1A also resumed OCR, ECAR, glucose uptake, and lactate production (Supplementary Fig. 8c and 8d) and restored TCA related metabolites and long-chain acylcarnitines (Supplementary Fig. 8e and 8f) and fatty acid synthesis (Supplementary Fig. 8g and 8h). In contrast, in LINC00842-overexpressing PDAC cells, overexpression of wild-type PPARGC1A suppressed cell proliferation, migration, and invasion (Supplementary Fig. 9a and 9b); overexpression of wild-type PPARGC1A also resumed OCR, ECAR, glucose uptake, and lactate production (Supplementary Fig. 9c and 9d) but suppressed fatty acid synthesis (Supplementary Fig. 9e and 9f). Moreover, overexpression of mutant PPARGC1A had the effects similar to wild-type PPARGC1A. In addition, we found that in PDAC cells with PPARGC1A overexpression, upregulation of LINC00842 significantly resumed OCR, ECAR, glucose uptake, lactate production, and fatty acid synthesis (Supplementary Fig. 10a–d). Together, these results clearly demonstrate that overexpressed LINC00842 prevents acetylated PGC-1α from deacetylation by SIRT1, resulting in metabolic remodeling of PDAC cells.

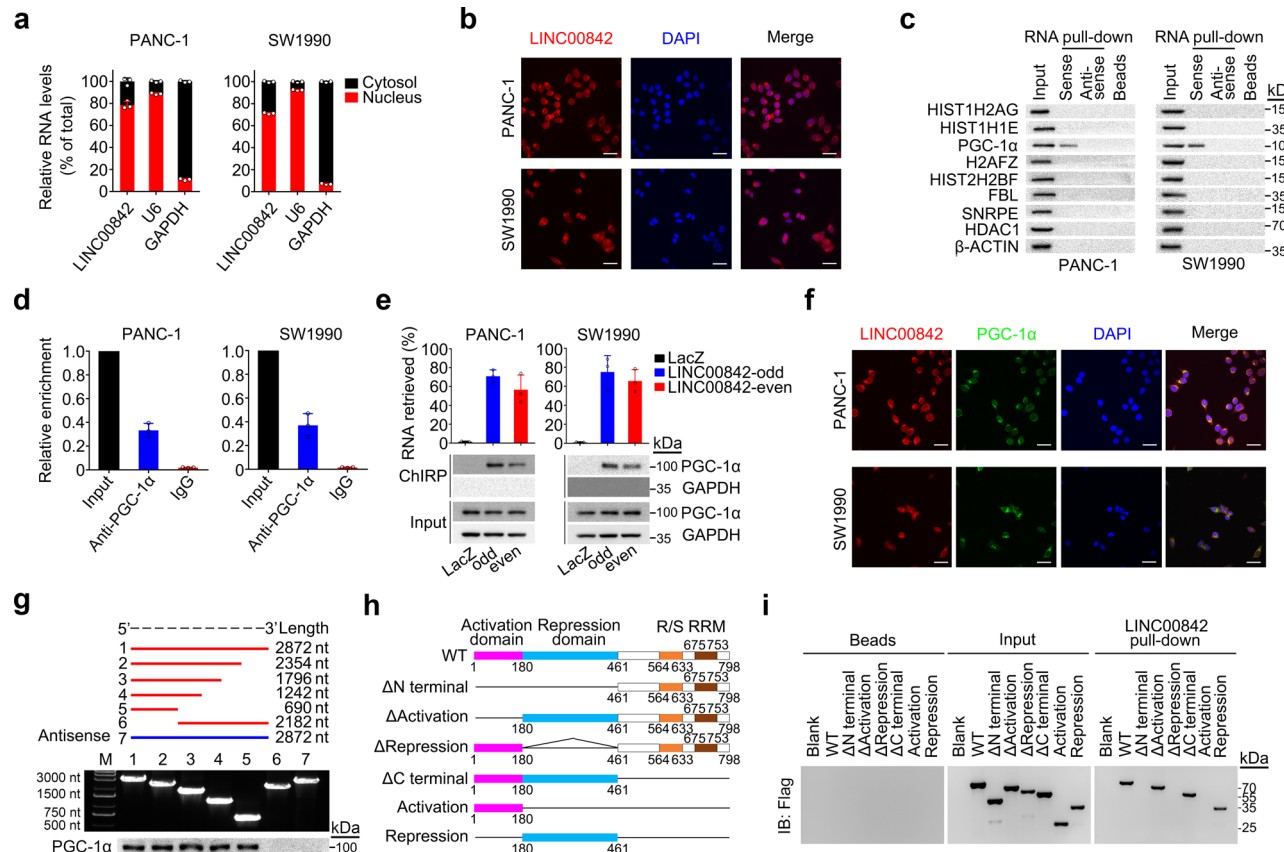

**Fig. 4 LINC00842 directly interacts with PGC-1α in PADC cells. a, b** Subcellular localization analysis by PCR (**a**) or RNA FISH (**b**) shows *LINC00842* is mainly located in the nuclei. *U6* and *GAPDH* were served as nuclear and cytoplasmic markers and shown are mean ± SD from 3 independent experiments (**a**). Scale bar, 30 μm (**b**). **c** Western blot analysis of proteomic screening suggested 8 potential *LINC00842*-binding proteins obtained from RNA pull-down assays with *LINC00842* or its antisense. **d** Association of PGC-1α with *LINC00842* determined by RNA immunoprecipitation (RIP) assays. Data represent relative enrichment (mean ± SD) to input from 3 independent experiments. IgG was used as the negative control. **e** Results of chromatin isolation by RNA purification (ChIRP) assays using *LINC00842*-odd, -even or *LacZ* (negative control) antisense probe sets. Upper panels are RNA retrieval rate (mean ± SD from 3 assays) and lower panels are immunoblot of PGC-1α and GAPDH (negative control) in ChIRP and input. **f** Immunofluorescence assays show co-localization of *LINC00842* (red) and PGC-1α (green) predominately in the nuclei. Scale bar, 30 μm. **g** Truncation mapping of *LINC00842* PGC-1α binding domain. Top panel: diagrams of *LINC00842* full-length and truncated fragments. Middle: RNA sizes of in vitro transcribed *LINC00842* full-length and truncated fragments. Bottom: immunoblot analysis of PGC-1α pulled down by different *LINC00842* fragments. **h** Schematic diagram of Flag-tagged PGC-1α and its truncated forms used in *LINC00842* pull-down assays. **i** Immunoblot analysis of Flag-tagged wild-type (WT) PGC-1α and its truncated forms retrieved by in vitro transcribed biotinylated *LINC00842*.

**LINC00842 expression in cells can be enhanced by exposure to high glucose through YY1.** We were interested in exploring why *LINC00842* is overexpressed in PDAC. With the above results and knowledge that diabetes mellitus is a risk factor for PDAC, we conjectured that high glucose concentration may promote aberrant *LINC00842* expression. We incubated immortalized human pancreatic duct epithelial cells (HPDE6-C7) and PDAC cells (PANC-1 and SW1990) with glucose at different concentrations (5 mM represents the physiological glucose concentration and 25 mM simulate hyperglycemia condition[28–33]) and then determined the cellular levels of *LINC00842*. Interestingly, we found that cells cultured with 25 mM of glucose had significantly higher *LINC00842* levels than those cultured with 5 mM of glucose (Fig. 6a and Supplementary Fig. 11a); however, the levels were reduced by 75% in 4 h after glucose concentration in the medium was switched from 25 to 5 mM (Fig. 6b) while no significant changes were observed at this time when cells were kept in 25 mM of glucose. We then selected 4 h as a time point for further experiments using the medium with different glucose concentrations. Subsequent assays showed that cells exposed to glucose had significantly enhanced interaction between *LINC00842*

and PGC-1α or its Repression domain in a dose-dependent manner (Fig. 6c and d). Acetylated PGC-1α was decreased when glucose concentration was switched from 25 to 5 mM (Fig. 6e) and this reduction could be reversed by *LINC00842* overexpression (Fig. 6f). Further results showed that decreased acetylated PGC-1α level in cells exposed to 5 mM glucose corresponded to increased interaction of PGC-1α with SIRT1; however, *LINC00842* overexpression reduced the interaction of two proteins, despite high glucose concentration (Fig. 6h). In contrast, *LINC00842* depletion decreased acetylated PGC-1α level even in cells exposed to 25 mM of glucose (Fig. 6g, i and Supplementary Fig. 11b). In parallel, the malignant phenotypes were significantly decreased in PDAC cells exposed to low glucose (5 mM) compared with those exposed to high glucose (25 mM). However, *LINC00842* overexpression significantly increased the malignant phenotypes even in PDAC cells exposed to low glucose (Supplementary Fig. 12a and 12b); *LINC00842* silence significantly repressed the malignant phenotypes in cells exposed to high glucose (Supplementary Fig. 12c and 12d). For the metabolic phenotypes, cells exposed to low glucose (5 mM) had increased OCR and decreased ECAR compared with those exposed to high

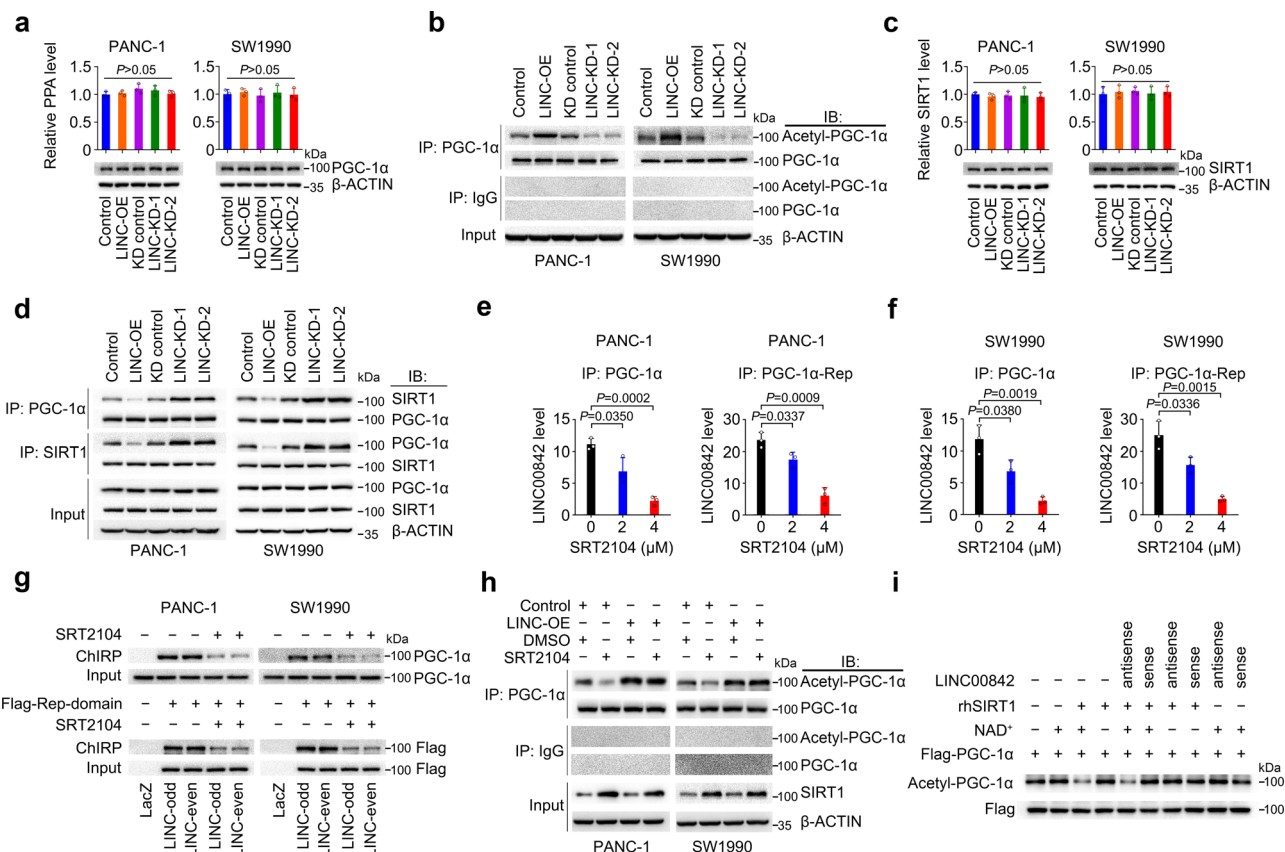

**Fig. 5 *LINC00842* reduces deacetylation of PGC-1α by SIRT1 in PDAC cells. a** *LINC00842* overexpression (LINC-OE) or silence (LINC-KD) did not affect *PPARGC1A* (PPA) mRNA and protein levels. **b** *LINC00842* expression change altered acetylated PGC-1α (acetyl-PGC-1α) levels. **c** *LINC00842* expression change did not affect *SIRT1* mRNA and protein levels. **d** Reciprocal immunoprecipitation assays show inhibitory effect of *LINC00842* on the interaction of PGC-1α with SIRT1. **e, f** Effect of SIRT1 agonist SRT2104 on the interaction of PGC-1α or its Repression domain (PGC-1α-Rep) with *LINC00842*. The level of *LINC00842* RNA in RNA immunoprecipitation product decreased in a SRT2104 dose-dependent manner. **g** Results of chromatin isolation by RNA purification (ChIRP) assays using *LINC00842*-odd, -even or *LacZ* (negative control) antisense probe sets. SRT2104 treatment reduced the interaction of PGC-1α or its Repression domain with *LINC00842*. **h** Immunoprecipitation and immunoblotting assays show inhibitory effect of *LINC00842* on PGC-1α deacetylation by SRT2104 (4 μM). **i** Effects of *LINC00842* sense or antisense on in vitro PGC-1α-Lys deacetylation by recombinant human SIRT1 (rhSIRT1) in the presence or absence of NAD+, showing only *LINC00842* sense inhibited the deacetylation. Flag was blotted as a loading control. The immunoblots in (**g–i**) are representative results from 3 independent experiments with similar results. The results shown in (**a**), (**c**), (**e**), and (**f**) are mean ± SD from 3 independent experiments. The *P* values were determined by Student's *t*-test (two-tailed).

glucose (25 mM); overexpressing *LINC00842* could counteract the effects of lower glucose (Supplementary Fig. 12e). In addition, *LINC00842* overexpression also significantly increased glucose uptake and lactate production in PDAC cells even exposed to low glucose (Supplementary Fig. 12f and 12g). In contrary, *LINC00842* silence counteracted OCR, ECAR, glucose uptake, and lactate production in cells even exposed to high glucose (Supplementary Fig. 12h–j).

Since high glucose apparently enhances *LINC00842* expression, we presumed that this effect might be mediated by *trans*-element(s). To test this presumption, we performed in silico analysis using the JASPAR and AnimalTFDB databases to search for *cis*-element(s) in the promoter region of *LINC00842* (from −2000 base pair to the transcription start site) and also looked at TCGA database to search for the transcription factors (TFs) positively correlated with *LINC00842* levels ($r > 0.3$, $P \leq 0.05$). By overlapping the results from these approaches, we identified 6 potential TFs for *LINC00842* expression (Fig. 6j and Supplementary Data 2). Further experiments with gene silencing showed that among the 6 TFs, only silencing *YY1* expression significantly inhibited *LINC00842* levels in cells exposed to 25 mM of glucose (Fig. 6k and Supplementary

Fig. 11c). In silico analysis showed a *cis*-element for YY1 located within −1807 to −1796 base pairs upstream of the *LINC00842* transcriptional start site (Fig. 6l) and chromatin immunoprecipitation assays verified the specific binding of YY1 to the *LINC00842* promoter (Fig. 6m). Reporter gene assays showed that mutations of the *cis*-element substantially abolished the promoter activity and transcription induction by high glucose (Fig. 6n and o). It has been shown that YY1 may interact with PGC-1α dependent on mTOR[34]. However, we found that *YY1* overexpression in PDAC cells significantly increased *LINC00842* level but *PPARGC1A* knockdown did not have this effect (Supplementary Fig. 13a); in parallel, *YY1* depletion significantly decreased *LINC00842* level but *PPARGC1A* overexpression did not have this effect (Supplementary Fig. 13b). Furthermore, reporter gene assays showed that disturbing *PPARGC1A* expression did not alter YY1-mediated expression of reporter gene with *LINC00842* promoter (Supplementary Fig. 13c and 13d). These results indicated that it is likely YY1 that is responsible for high glucose-induced *LINC00842* expression. Indeed, we found that PDAC cells exposed to high glucose had substantially higher YY1 levels than had cells exposed to low glucose (Fig. 6p), consistent with

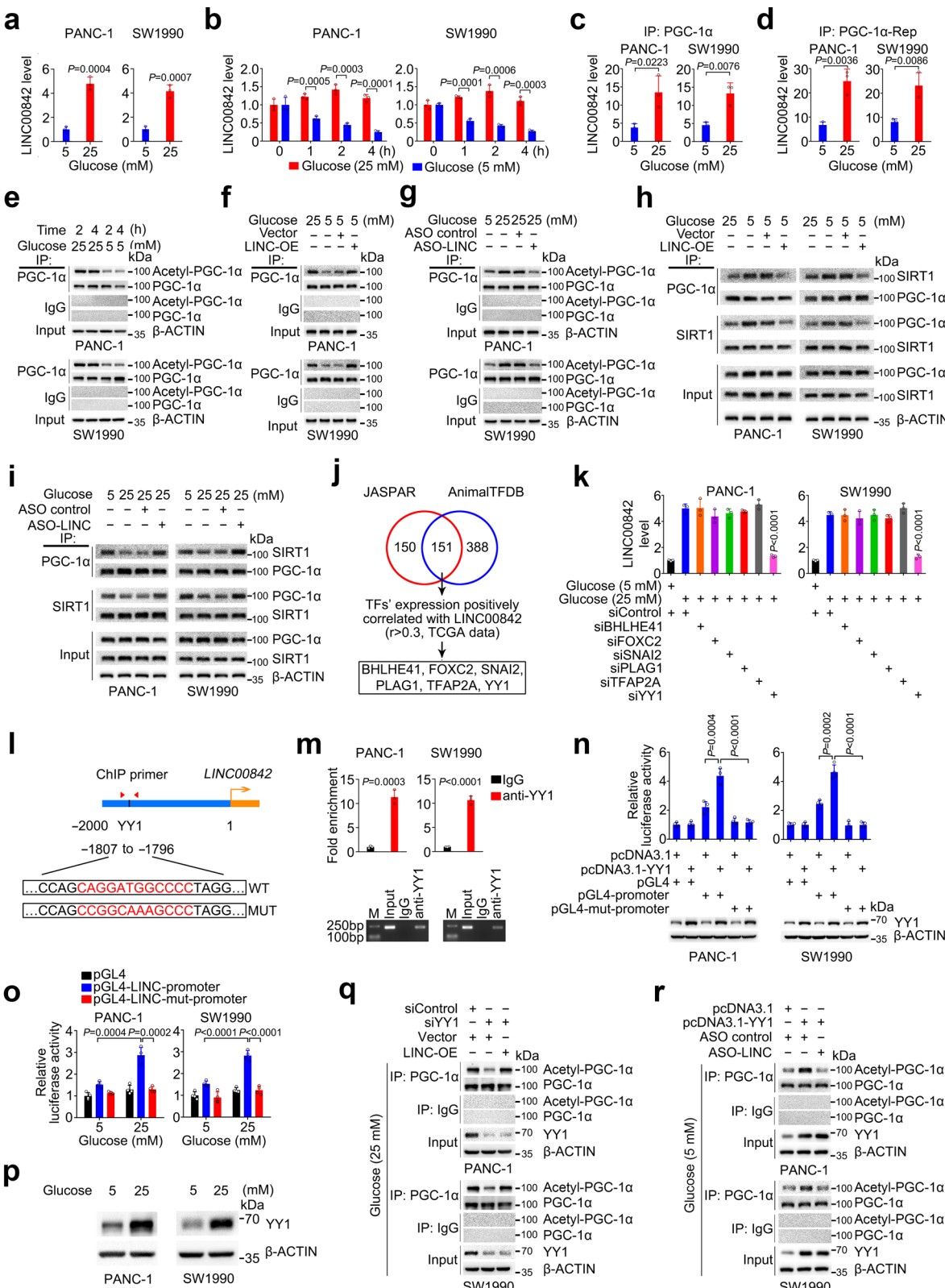

previous finding in diabetic nephropathy[35]. In addition, we found that the effect of *YY1* silencing on PGC-1α acetylation was rescued by *LINC00842* overexpression in cells exposed to high glucose while the effect of YY1 overexpression on PGC-1α acetylation was reversed by *LINC00842* silencing in cells exposed to low glucose concentration (Fig. 6q and r), demonstrating that *LINC00842* acts downstream of YY1.

**LINC00842 is a potential therapeutic target for PDAC.** We finally examined the levels of *LINC00842* RNA and some critical proteins (YY1, FASN, and acetylated PGC-1α) in 30 paired PDAC and adjacent normal samples and analyzed their correlations. The results indicated that PDAC tumors had significantly higher levels of *LINC00842* RNA and the 3 proteins than their adjacent normal tissues (Fig. 7a and b). Furthermore, *LINC00842*

**Fig. 6 High glucose induces *LINC00842* expression via transcription factor YY1 in PDAC cells. a** *LINC00842* levels in cells cultured with 5 or 25 mM of glucose (*n* = 3). **b** *LINC00842* levels in cells at different times after switch of glucose dose in culture medium (*n* = 3). **c, d** Association of PGC-1α (**c**, *n* = 3) or its Repression domain (PGC-1α-Rep, **d**, *n* = 3) with *LINC00842* determined by RNA immunoprecipitation (RIP) assays in cells cultured with different glucose doses. **e** Immunoprecipitation and immunoblot analysis of acetyl-PGC-1α levels in cells cultured with 25 mM of glucose at 2 and 4 h and after switch to 5 mM of glucose for 2 and 4 h. **f, g** Immunoprecipitation and immunoblot analysis of acetyl-PGC-1α levels in cells with *LINC00842* overexpression (**f**) or knockdown (**g**) exposed to indicated glucose doses for 4 h. **h, i** Co-immunoprecipitation analysis of PGC-1α and SIRT1 interaction in cells with *LINC00842* overexpression (**h**) or knockdown (**i**) exposed to indicated glucose doses for 4 h. **j** Prediction of potential transcription factors in the *LINC00842* promoter region. **k** Relative *LINC00842* levels in cells with or without knockdown of each of the 6 transcription factors indicated in (**j**) (*n* = 3). **l** Schema of the putative YY1 binding site in the promoter of *LINC00842* gene and the primers used for chromatin immunoprecipitation (ChIP) analysis. The consensus and mutant sequences for YY1 binding were boxed. **m** ChIP analysis with anti-YY1 antibodies or IgG control. Upper panel shows qPCR results (*n* = 3) and lower panel are images of agarose gel electrophoresis of the qPCR products. **n** Luciferase reporter assays in cells co-transfected with the indicated plasmids for 48 h (*n* = 4). **o** Luciferase reporter assays in cells cultured in 5 mM of glucose and then switched to 25 mM of glucose for 4 h (*n* = 4). **p** Immunoblot analysis of YY1 level in cells exposed to 5 or 25 mM of glucose. **q** Immunoprecipitation and immunoblot analysis of acetyl-PGC-1α levels in cells transfected with the indicated plasmids or siRNAs and exposed to 25 mM glucose. **r** Immunoprecipitation and immunoblot analysis of acetyl-PGC-1α levels in cells transfected with the indicated plasmids or antisense oligonucleotide (ASO) of *LINC00842* and exposed to 5 mM glucose. The results in (**a–d**), (**k**), (**m-o**) are mean ± SD from at least 3 independent experiments. The *P* values were determined by Student's *t*-test (two-tailed).

RNA levels as well as YY1, FASN, and acetylated PGC-1α protein levels were significantly and positively correlated with each other in PDAC tumor and normal tissues (Fig. 7c, d, Supplementary Fig. 14a and 14b). Kaplan–Meier estimation showed that patients with high *FASN* levels (≥median) had shorter survival time than patients with low *FASN* levels (<median) in both Cohort 1 and Cohort 2 (Supplementary Fig. 14c). Since *LINC00842* plays a critical role in metabolic remodeling in PDAC, we created mouse orthotopic implantation and PDX models and performed experimental therapy with Orlistat, a FASN inhibitor[36], and in vivo-optimized ASO-LINC00842, a *LINC00842* inhibitor. We found that treatment with ASO-LINC00842 or Orlistat in mice bearing orthotopically implanted tumor derived from *LINC00842*-overexpressing PDAC cells significantly reduced tumor burden and increased survival time compared with controls and combined treatment of these two reagents had better benefits than treatment with each reagent alone (Fig. 7e–g). In mouse PDX models, treatment with ASO-LINC00842 and Orlistat (Fig. 7h) significantly inhibited PDAC growth and the two agents exhibited synergistic effect (Fig. 7i and Supplementary Fig. 14d). In addition, we found that treatment of ASO-LINC00842 significantly repressed *LINC00842* levels (Fig. 7j), increased the PGC-1α and SIRT1 interaction (Supplementary Fig. 14e), decreased PGC-1α acetylation (Fig. 7k and Supplementary Fig. 14f), and suppressed FASN expression in PDXs. A synergistic effect of combined treatment of ASO-LINC00842 and Orlistat on FASN inhibition was also seen in tumors (Fig. 7l). Together, these results indicate that *LINC00842* and its relevant metabolic remodeling may be druggable targets for PDAC therapy.

## Discussion

In the present study, we have identified a long intergenic non-coding RNA, *LINC00842*, whose aberrant overexpression is associated with PDAC tumor growth, invasiveness, and shorter survival time in patients. We have found that high glucose level provokes cells to express *LINC00842* that is likely mediated by transcription factor YY1. Overexpressed *LINC00842* may directly interact with acetylated PGC-1α, an important transcription co-regulator in the metabolic pathways, and blocks SIRT1 to deacetylate acetylated PGC-1α, resulting in metabolic remodeling, i.e., enhancing anabolism but reducing catabolism in cancer cells (Fig. 7m). Finally, we have demonstrated that targeting *LINC00842* or the downstream FASN significantly suppressed the growth of PDAC tumors derived from PDAC cell lines or patient tumors in mice. These results shed light on a molecular mechanism underlying the oncogenic action of an aberrantly expressed lincRNA in PADC and provide a potential druggable target for PDAC treatment.

*LINC00842* is a known lincRNA expressed in most of human tissues with a level in normal pancreatic tissues being comparatively low[37] (present study); however, its levels are significantly increased in PDAC, especially in advanced tumors, indicating that this lincRNA indeed plays an important role in PDAC development and progression. Interestingly, we have found that high concentration of glucose can induce *LINC00842* expression in both normal pancreatic cells and PDAC cells, suggesting an epigenetic regulation role of glucose through this lincRNA. These findings might provide an explanation for why patients with abnormal glucose metabolism, i.e., hyperglycemia, are more susceptible to PDAC[38]. The regulatory effects of glucose on the expression of some genes and protein posttranslational modifications have been documented[39]. For example, glucose (but not its osmotic pressure) has been reported to induce YY1 expression in diabetic nephropathy-induced renal fibrosis[35], which is consistent with our result showing that YY1 is involved in *LINC00842* induction in PDAC cells exposed to high glucose concentration. These biologically plausible findings may open an avenue to investigating the association between diabetes mellitus and PDAC. Nevertheless, one previous study has shown that *LINC00842* was induced by TGF-β stimulation via the TGF-β/SMAD pathway in TGF-β-responsive cell lines, such as human hepatocarcinoma Huh7 and lung adenocarcinoma A549 cells, and then targeted SMAD3 to promote epithelial–mesenchymal transition[40]. However, in our experiment setting in PDAC cells, we did not observe such an effect but found a unique function of *LINC00842* in metabolic remodeling. These findings suggest that *LINC00842* may have different functions in different types of cancer.

Transcriptional coregulators PGC-1α regulates the expression of genes involved in mitochondrial oxidative phosphorylation and energy homeostasis[41–44]. Aberrant expression of this regulator has been associated with cancer initiation and development. However, some studies showed PGC-1α promoting cancer metastasis[45] while others showed suppressing cancer metastasis[46,47], suggesting that the role of PGC-1α in cancer is complicated. It has been shown that lncRNA *Tug1* may interact with PGC-1α and recruit the protein to its own gene promoter to regulate its expression[13]. However, in the present study, we have demonstrated that *LINC00842* does not disturb PGC-1α expression; instead, it binds to acetylated PGC-1α and prevents the protein from deacetylation by SIRT1. It is well known that only non-acetylated PGC-1α can activate mitochondrial oxidative processes and attenuates fatty acid biogenesis; thus, PGC-1α

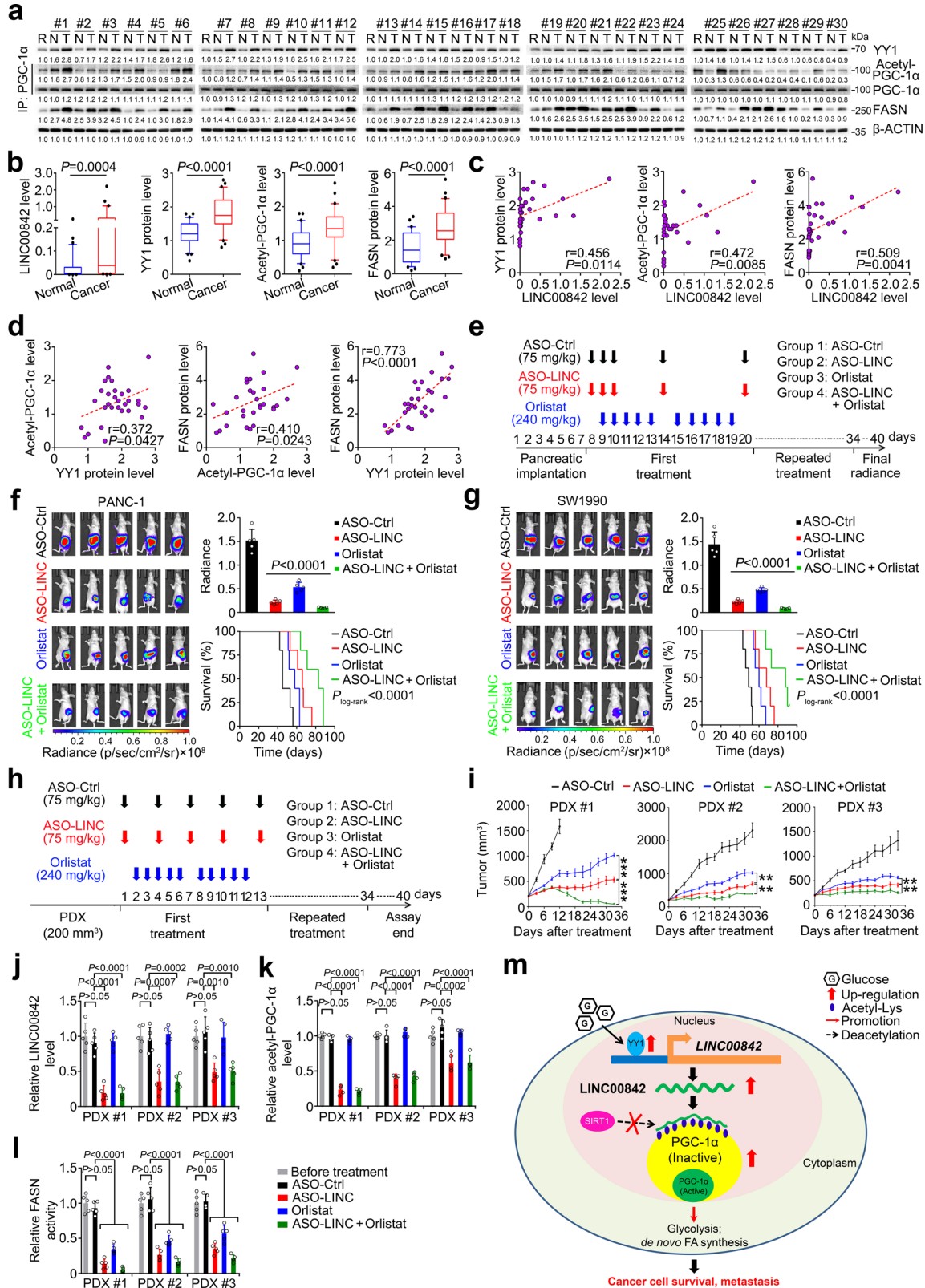

should play an important role as a tumor suppressor in the development and progression of PDAC.

Because metabolic remodeling is one of the hallmarks of cancer cells, it represents an attractive therapeutic target for cancer treatment[48,49]. In the present study, we have found that over-expression of *LINC00842* in PDAC provokes fatty acid synthesis, which is attributable to the upregulation of FASN, the key enzyme

responsible for the terminal catalytic step in fatty acid synthesis. FASN is highly expressed in many types of cancer and thus has been a suggestive therapeutic target[50]. Indeed, we have observed that ASO-LINC00842, which inhibits *LINC00842*, or Orlistat, which inhibits FASN activity, and combination of these two agents can significantly and synergistically repress PDAC growth and tumor burden in mouse xenograft and PDX models. These

**Fig. 7 LINC00842 is a potential therapeutic target for PDAC. a** Immunoprecipitation and immunoblot analysis of YY1, acetyl-PGC-1α, and FASN in PDAC and paired non-tumor tissues ($n = 30$) from one experiment. **b** Levels of *LINC00842* RNA and YY1, FASN, and acetyl-PGC-1α proteins in PDAC and paired non-tumor tissues ($n = 30$). RNA levels were determined with qPCR while protein levels were determined with western blotting and quantified using Image J. Data are shown in box plots; the lines in the middle of the box are median and the upper and lower lines indicate 25th and 75th percentiles. The *P* values were determined by Wilcoxon test (two-tailed). **c**, **d** Spearman's correlations between levels of *LINC00842* and YY1, acetyl-PGC-1α or FASN (**c**) and between YY1, acetyl-PGC-1α, and FASN (**d**) in PDAC tissues ($n = 30$). **e** Timeline schematic for treatment of mice carrying xenograft in the pancreas with antisense oligonucleotide (ASO)-control (Ctrl), ASO-LINC00842, Orlistat or combination of ASO-LINC00842 and Orlistat. Arrows indicate different treatment time points. **f**, **g** Bioluminescence imaging (left panel), quantitative fluorescent intensities (upper right panel), and survival time (lower right panel) of mice with different treatment. Fluorescent intensity data represent mean ± SD from 5 mice in each group; The *P* values were determined by Student's *t*-test (two-tailed). **h** Timeline schematic for treatment of mice carrying patient-derived tumor xenograft (PDX). Arrows indicate different treatment time points. **i** ASO-LINC00842 and Orlistat treatment significantly inhibited PDX growth in mice. Data in each time point represent mean ± SEM from 5 mice. **$P < 0.01$ and ***$P < 0.001$ of Student's *t*-test (two-tailed. No adjustments were made for multiple comparisons). **j–l** *LINC00842* RNA (**j**), acetyl-PGC-1α (**k**), and FASN (**l**) protein or activity levels in PDX of mice before and after treatment. Data are mean ± SD from 5 mice; The *P* values in (**j–l**) were determined by Student's *t*-test (two-tailed). **m** Proposed acting model for *LINC00842* that regulates metabolic reprogramming in PDAC.

results not only verify the molecular mechanism underlying the role of *LINC00842* in PDAC development and progression, but also suggest that decreasing *LINC00842* and FASN expression or inhibiting their activity may represent new therapeutic targets for PDAC. Since Orlistat has currently been used in clinical trials for weight loss[36], it would be interesting and warrant to test whether it is of effectiveness in treatment of PDAC.

We acknowledge some limitations in the current study. Although we focused simply on glucose-induced *LINC00842* logically in the context of our study, it would be interesting to globally profile the entire ncRNAs in PDAC cells exposed to high glucose concentration. In addition, the underlying mechanism for how glucose induces YY1 expression remains to be uncovered. Finally, further studies are needed to clarify whether the effect of *LINC00842* on metabolic remodeling is PDAC specific or also presents in other types of cancer.

In summary, we have identified a metabolism-remodeling ncRNA *LINC00842* in PDAC that can be induced by high glucose, an etiological risk factor for PDAC development. *LINC00842* promotes proliferation and invasiveness of PDAC cells most likely by switching the cellular mitochondrial oxidative catabolic processes to fatty acid synthesis. Decreasing *LINC00842* or inhibiting its downstream fatty acid synthesis activity has achieved significant therapeutic effects in vitro and in vivo in xenograft and PDX models. These results shed light on the important role of ncRNA in cancer development and progression and may provide a fundamental for developing more target PDAC therapies.

## Methods

**Study subjects**. Surgically removed PDAC and the corresponding adjacent normal tissue samples ($n = 227$) were obtained from Sun Yat-sen Memorial Hospital, Sun Yat-sen University Cancer Center (Guangzhou, China, $n = 158$; cohort 1) and Cancer Hospital, Chinese Academy of Medical Sciences (Beijing, China, $n = 69$; cohort 2) between 2005 and 2018 (Supplementary Table 3). The characteristics and clinical data of patients were acquired from medical records. Survival time was measured from the date of diagnosis to the date of last follow-up or death. The last date of follow-up was 30th December 2018, and the median follow-up time was 39 months. Patients alive on the last follow-up data were censored. Informed consent was obtained from all patients, and this study was approved by the Institutional Review Board of Sun Yat-sen Memorial Hospital, Sun Yat-sen University Cancer Center and Cancer Hospital, Chinese Academy of Medical Sciences.

**Analysis of public datasets**. RNA-sequencing data of 177 PDAC patients were retrieved from TCGA PAAD sample level 3 access (http://gdac.broadinstitute.org, version 2016_01_28; PMID: 28810144) and lncRNAs were annotated by Ensembl (PMID: 31691826). After filtering out lncRNAs with an abundance of 0 in more than half of the samples, 1908 lncRNAs were used for survival analysis through the Wald test. After adjusting to control the false discovery rate (FDR) by Benjamini and Hochberg approach, the expression levels of 42 lncRNAs were associated with survival time in PDAC patients, and 10 top lncRNAs ranking by the FDR-value were selected for further investigation (Fig. 1a and Supplementary Data 1). We also downloaded RNA array data and clinical information of PAAD patients from

ICGC (https://dcc.icgc.org/releases/release_28/Projects/PACA-AU) and used for survival analysis. For placental mammal conservation analysis, PhastCons from UCSC (http://genome.ucsc.edu/) was used to analyze *LINC00842* and its nearby coding gene. Prediction of the transcription factor (TF) binding sites in the *LINC00842* gene promoter region (−2000 bp to transcription start site) were performed using JASPAR (PMID: 31701148) and AnimalTFDB (PMID: 30204897) and TFs-*LINC00842* co-expression analysis was integrated. Only TFs with binding site relative score >0.8 and positive correlation coefficient $r \geq 0.3$ were considered, thus resulting in 6 potential TFs including BHLHE41, FOXC2, SNAI2, PLAG1, TFAP2A, and YY1. Co-expressions of the TF and *LINC00842* were evaluated with Spearman's correlation coefficient based on TCGA PAAD RNA-sequencing data (Supplementary Data 2). Prediction of the interaction between 5′-end of *LINC00842* (nucleotides 1–690) and PGC-1α were performed using PRIdictor (http://www.rna-society.org/raid/PRIdictor.html) and followed by RPISeq (http://pridb.gdcb.iastate.edu/RPISeq/index.html), the prediction based on RF (Random Forest) classifiers >0.5 were considered for further validation. To explore the protein-coding potential of *LINC00842*, we compared ribosome-protected fragments of *LINC00842* with those of canonical coding and non-coding RNAs. Ribo-seq data were retrieved from TranslatomeDB database[23].

**Cell lines and cell culture**. Human PDAC cell lines PANC-1 and SW1990 and human embryonic kidney cell line 293T were purchased from the Cell Bank of Type Culture Collection of the Chinese Academy of Sciences Shanghai Institute of Biochemistry and Cell Biology. Human immortalized pancreatic duct epithelial cell line HPDE6-C7 was purchased from Biotechnology Company. Cells were cultured in DMEM with 10% FBS, and 5 or 25 mM glucose was supplied where it was necessary. Cells passaged for <6 months were authenticated by DNA fingerprinting analysis using short-tandem repeat (STR) markers.

**Northern blot assays**. Total RNA (50 μg) from cells was separated on agarose gels containing formaldehyde and transferred to Biodyne Nylon Membrane (Pall). After immobilization and pre-hybridization of RNA in DIG Easy Hybrid buffer (Roche), the membrane was hybridized overnight at 68 °C in DIG Easy Hybrid buffer containing the denatured digoxigenin-labeled *LINC00842* probes (BersinBio, Supplementary Table 4). After washing, the membrane was incubated with anti-digoxigenin-AP and detected using an Odyssey infrared scanner (Li-Cor, Lincoln, Supplementary Fig. 1c).

**RNA extraction and qRT-PCR analysis**. Total RNA from cell lines and pancreatic tissue specimens was extracted using TRIzol reagent (Invitrogen). First-strand cDNA was synthesized using the RevertAid First Strand cDNA Synthesis Kit (Thermo) with random primers. Relative RNA level was determined by RT-qPCR on a Light Cycler 480 II using the SYBR Green method. The sequences of the gene-specific primers are listed in Supplementary Table 4. β-ACTIN was used as an internal control for quantification of *LINC00842* and the mRNA levels of other genes. Three biological replicates were set up for each experiment. The relative expression level of RNAs was calculated using the comparative Ct method.

**RNA sequencing and KEGG pathway analysis**. Total RNA extracted from PANC-1 and SW1990 cells with or without *LINC00842* silenced was subjected to RNA sequencing (RiboBio, Illumina HiSeq2500). The raw and processed sequencing data have been submitted to the Gene Expression Omnibus (accession number GSE150216). For Kyoto Encyclopedia of Genes and Genomes (KEGG) analysis, differentially expressed protein-coding genes ($|\log_2(\text{Fold change})| > 1$) were included, and $P < 0.05$ was considered significant for biological process enrichment.

**Measurement of absolute LINC00842 copy number per cell**. We determined the accurate *LINC00842* copy number per cell by using absolute qPCR[51]. For

standard curve generation, we made a series of concentration gradient of in vitro transcribed full-length *LINC00842* sense strand RNA, then first-strand cDNA was synthesized using the RevertAid First Strand cDNA Synthesis Kit (Thermo) and performed qPCR. For sample test, total RNA from $3 \times 10^6$ PDAC cells was extracted using TRIzol reagent (Invitrogen). First-strand cDNA was synthesized and then perform qPCR examination. Each Ct value corresponds to one RNA concentration on the standard curve. The exact copy number of *LINC00842* per cell was calculated based on *LINC00842* molecular weight and cell count (Supplementary Fig. 1d).

**Plasmid construction and transient transfection.** The small guide RNA (sgRNA) targeting the transcription start site of *LINC00842* or control sgRNA was designed and subcloned into the pLenti-U6-spgRNA v2.0-CMV-sfGFP-P2A-3Flag-spCas9 vector (H6825, Obio Technology; Supplementary Table 1). The CMV promoter-puro-3 × poly (A) segment was cloned into the donor vector to integrate into the *LINC00842* transcription start site with the designed sgRNA (AOV021, Obio Technology). The cDNA for the *YY1* (NM_003403.5), Flag-tagged *PPARGC1A* (NM_013261.5), truncated versions of Flag-tagged *PPARGC1A* and Flag-tagged acetylation sites mutant *PPARGC1A* were cloned into the pcDNA3.1 vector (Obio Technology). For the reporter gene assays, *LINC00842* promoter or its mutated form was cloned into the pGL4.10 vector (Obio Technology). All constructed plasmids were verified by DNA sequencing. Transient transfection of plasmids or small interfering RNAs (GenePharma, Supplementary Table 1) was performed with Lipofectamine 3000.

**Lentiviral production and transduction.** To construct a lentiviral vector expressing human *LINC00842* (NR_033957.2), full-length of *LINC00842* cDNA was commercially synthesized and subcloned into a lentiviral expression vector (H145, Obio Technology). Both control vector and recombinant plasmid were then transfected into 293T cells to produce lentivirus, which, respectively, infected PANC-1 and SW1990 cells. The supernatant was replaced with complete culture media after 24 h, and then the cells were selected with puromycin.

**Construction of *LINC00842*-silencing PDAC cells.** We generated *LINC00842*-silencing PDAC cells by using the CRISPR Cas9 system[52] (Supplementary Fig. 1f). Five μg of pLenti-U6-spgRNA v2.0-CMV-sfGFP-P2A-3Flag-spCas9 vector containing control sgRNA or sgRNA targeting the *LINC00842* transcription start site was co-transfected with 15 μg of CMV promoter-puro-3 × poly (A) stop cassette donor vector. After electroporation, 0.5 μg/ml puromycin was added; individual colonies were picked and expanded in 96-well plates. When the cell population in each colony was large enough, genomic DNA was extracted and genomic integration was validated by PCR (Supplementary Fig. 1g). After testing the expression of *LINC00842* in CRISPR-edited PDAC cells (Supplementary Fig. 1h), we chose two positive colonies named 'LINC00842-KD-1' and 'LINC00842-KD-2' for further investigations.

**Examination of cell malignant phenotypes.** PDAC cells (2000 per well) were seeded in 96-well plates with 100 μl of medium supplied with 5 or 25 mM glucose. Cell viability was measured using CCK-8 assays (Dojindo) at defined time of culture. Each experiment was performed with at least three replicates. For colony formation assays, 500 cells per well were seeded in 12-well plates and allowed to grow until visible colonies formed in complete growth medium, which was fixed with methanol, stained with crystal violet, and counted. Invasion assays were performed in Millicell chambers in triplicate. The 8-μm pore inserts were coated with 30 μg of Matrigel (BD Biosciences). Cells ($5 \times 10^4$) were added to the coated filters in serum-free medium. We supplied DMEM containing 20% FBS to the lower chambers as a chemoattractant. After 20 h at 37 °C in an incubator with 5% $CO_2$, cells migrated through the filters were fixed with methanol and stained with crystal violet. Cell numbers in 3 random fields were counted. The migration assay was constructed in a similar approach without Matrigel.

**Establishment of mouse xenograft models.** Female BALB/c nude mice aged 5 weeks (Beijing Vital River Laboratory Animal Technologies) were allowed to acclimate to local conditions for 1 week and maintained under a 12-h dark/12-h light cycle with adequate food and water. Animals were then randomly grouped ($n = 5$ per group) and subcutaneously injected with 0.1 ml of cell suspension containing $2 \times 10^6$ PDAC cells in the back flank. When a tumor was palpable, it was measured every week, and its volume was calculated using the formula volume = $0.5 \times$ length $\times$ width$^2$.

We also implanted luciferase-labeled PDAC cells ($2 \times 10^6$) to mouse pancreas by surgical injection. Tumor volume was monitored twice a week by bioluminescence imaging with a Living Image® system (Perkin Elmer), and the quantitative data were expressed as photon flux. The pancreas, lung, liver, and intestines of each mouse were removed when animal was scarified for further pathological examination.

For creating mouse PDX models, fresh PDAC obtained from three patients who underwent surgery without any chemotherapy or radiation therapy were propagated as subcutaneous tumors in 4-week-old NSG mice (F1). Xenografts from F1 mice were cut into small pieces and then implanted into other mice (F2). When

tumors grew up to about 1500 mm$^3$, they were excised and cut again into small pieces and transplanted to other mice (F3).

**Treatment of mice carrying xenografts.** Seven days after orthotopic implantation of PDAC cells, mice were divided into four groups ($n = 5$ per group) and treated, respectively, with: (a) antisense oligonucleotide (ASO)-Ctrl (75 mg/kg; RiboBio; Supplementary Table 1) dissolved in saline, i.v. injection every day for the first 3 days and then once a week; (b) Orlistat (240 mg/kg; Selleck) dissolved in 33% of ethanol/67% polyethylene glycol (v/v), i.p. injection every day with a break every 5 days; (c) ASO-LINC00842 (75 mg/kg; RiboBio; Supplementary Table 1), i.v. injection every day for the first 3 days and then once a week; and (d) combination of ASO-LINC00842 and Orlistat (Fig. 7e). Tumor burden of each mouse was monitored twice a week by bioluminescence imaging and survival time was recorded from the day of tumor implantation to the date of death.

When PDX grew up to about 200 mm$^3$, F3 mice were randomly divided into four groups ($n = 5$ per group) and treated with: (a) ASO-Ctrl, intra-tumor injection every 3 days; (b) Orlistat (240 mg/kg), i.p. injection every day with a break every 5 days; (c) ASO-LINC00842 (10 mg/kg), intra-tumor injection every 3 days; (d) combination of ASO-LINC00842 and Orlistat (Fig. 7h). Tumor volume calculated as $0.5 \times$ length $\times$ width$^2$ was monitored every 3 days. All animal handling and experimental procedures were approved by the Institutional Animal Care and Use Committee of Sun Yat-Sen University and performed in accordance with the relevant institutional and national guidelines.

**Determination of FASN activity.** FASN activity was determined with Fatty Acid Synthetase Activity Assay Kit (BC0555, Solarbio). Briefly, mouse PDX tissue was lysed on ice and then centrifuged at $21,130 \times g$ for 40 min in 4 °C. The supernatant was transferred to a tube and the FASN activity was evaluated according to the assay kit instruction.

**Glucose metabolism assays.** The oxygen consumption rate (OCR) was measured with a Seahorse XFe 24 Extracellular Flux Analyzer with Agilent Seahorse XF Cell Mito Stress Test Kit (103015-100, Agilent). Briefly, PDAC cells ($3 \times 10^4$ per well) were seeded in a XFe 24 plate with 10% FBS DMEM medium for 24 h. One hour before detection, the medium was changed to Seahorse XF Base Medium with 1 mM of pyruvate, 2 mM of glutamine, and a specified concentration of glucose. Mitochondrial stress was tested using the compounds of oligomycin (1 μM), carbonylcyanide-4(trifluoromethoxy)phenylhydrazone (1 μM), and rotenone and antimycin A (each 0.5 μM). Extracellular acidification rate (ECAR) was monitored based on the XF Glycolysis Stress Test protocol (103020-100, Agilent) on a Seahorse XFe 24 Extracellular Flux Analyzer using the compounds of oligomycin (1 μM), glucose (specified), and 2-deoxyglucose (50 mM). Glucose uptake and lactate production were monitored using Glucose Assay Kit and L-Lactate Assay Kit (ab65333 and ab65330, Abcam). Briefly, PDAC cells were cultured in DMEM with 5 or 25 mM of glucose and the medium was then collected and cleared by protein precipitation and neutralization followed by the measurement of glucose and lactate levels. Medium collected from cell-free plates served as baseline control for estimating glucose consumption and lactate production. The results were normalized to cell number in each plate determined at the time of medium collection.

**Global metabolite profiling.** PDAC cells ($5 \times 10^6$) were collected in 80% HPLC-grade methanol (4 methanol/1 water, v/v) and centrifuged. The supernatant was transferred to a tube and dried under vacuum. The precipitate was dissolved in 50 μl of 80% HPLC-grade methanol and protein level was determined. We applied ultrahigh-performance liquid chromatography coupled with a TSQ Quantiva mass spectrometer (HPLC-MS, Thermo Scientific) to globally profile the metabolites. Metabolite identification and data processing were accomplished by TraceFinder 3.2 (Thermo).

For acylcarnitine analysis, PDAC cells ($5 \times 10^6$) were collected in 80% HPLC-grade methanol, sonicated, and centrifuged. The supernatant (3 μl) was mixed with 100 μl of $^2$H-labeled carnitine standard working solution dissolve in 2 ml HPLC-grade methanol (Cambridge Isotope Laboratories, Inc.). The mixture was centrifuged and the supernatant was collected and dried with $N_2$ at 50 °C. The sample was resuspended in 60 μl of acetyl chloride/1-butanol mixture (1:9, v/v) and incubated at 65 °C for 20 min. The derived sample was dried with $N_2$ and dissolved in 100 μl of 80% acetonitrile before being subject to an AB Sciex 4000 QTrap system (AB Sciex, Framingham, MA, USA). Data acquisition was conducted with Analyst 1.6 (AB Sciex) and metabolite identification and data processing were accomplished using ChemoView 2.0.2 (AB Sciex).

For $^{13}$C-U$_6$-glucose metabolism analysis, PDAC cells seed for 24 h were cultured in medium with $^{13}$C-U$_6$-glucose (25 mM) for another 48 h. Then the medium was removed and cells were exposed to $^{13}$C-U$_6$-glucose again for 2 h. The sample prepared as mentioned above was subject to HPLC-MS analysis. $^{13}$C-U$_6$-glucose tracing was also accomplished using gas chromatography-mass spectrometry (GC-MS, Thermo Fisher Scientific Trace 1300). Cells exposed to $^{13}$C-U$_6$-glucose were collected in extractant consisting of HPLC-grade methanol, tert-Butyl methyl ether (Sigma-Aldrich), and $H_2O$ (1/2/1, v/v/v) and extracted by vortex for 15 min. After centrifugation, both ether and methanol layers were, respectively, collected for analysis of fatty acid flux and TCA intermediate

metabolite flux. The ether extract was dried by $N_2$ and derived in 2% $H_2SO_4$/methanol solution (1/49, v/v) for 1 h at 50 °C. The derivatives extracted by saturated NaCl solution and hexane were dissolved in hexane and subject to GC-MS (Thermo Fisher). The methanol extract was dried under vacuum and derived in 2% (w/v) methoxyamine hydrochloride in pyridine for 1 h at 37 °C, followed by silylation in 30 μl of *N*-methyl-*N*-(tert-butyldimethylsilyl)trifluoroacetamide plus 1% of *tert*-butyldimethylchlorosilane (Regis Technologies) for 30 min at 45 °C. The resultant derivatives were subject to GC-MS.

**Determination of *LINC00842* in subcellular fractionations**. The Cytoplasmic & Nuclear RNA Purification Kit (NorgenBiotek Corp) was used to prepared cytoplasmic and nuclear fractions. RNA extracted from the cytoplasmic and nuclear fractions was determined by qPCR using *GAPDH* and *U6* as cytoplasmic and nuclear control.

**Analysis of cellular lipid by Nile red staining**. Cellular lipid was analyzed using Nile red staining assays. Briefly, PDAC cells were washed with PBS, fixed with 4% paraformaldehyde solution, and stained with Nile red solution for 8 min at 37 °C in dark to visualize lipid droplets in cells. Cell nuclei were counterstained with DAPI. Images were obtained with an Olympus FV1000 confocal microscope (Olympus).

**Analysis of citrate in subcellular fractionations**. Cultured cells were washed and collected in PBS. After centrifugation, the pellet was resuspended in acidic extraction reagent and grinded in tissue grinder on ice. Sample was centrifuged at $600 \times g$ for 5 min at 4 °C and the supernatant was transferred to a tube and centrifuged again at $11,000 \times g$ to isolate the cytosol (supernatant) and mitochondria (pellet). The pellet was suspended in acidic extraction reagent and sonicated on ice. After centrifugation, the supernatant was collected. The citrate levels in cell cytosol and mitochondria were determined with the Mitochondrial Citric Acid (MCA) Content Assay Kit (BC2170, Solarbio). The result was normalized by the cell numbers.

**Analysis of co-localization of *LINC00842* and PGC-1α**. Co-localization of *LINC00842* and PGC-1α protein was analyzed by RNA FISH performed with a lncRNA FISH Kit (RiboBio) and immunofluorescence staining of PGC-1α antibody. Briefly, PDAC cells were fixed and permeabilized in PBS containing 0.5% of Triton X-100. Hybridization with the FISH probes designed by RiboBio was carried out overnight in a humidified chamber at 37 °C in dark. 4′,6-diamidino-2-phenylindole and Cy3 channels were used to detect the signals. Immunofluorescence staining of PGC-1α was performed using PGC-1α antibody (1:200, NBP1-04676, Novus Biologicals, verification of antibody specificity was shown in Supplementary Fig. 15a) and Alexa Fluor® 488 conjugated secondary antibody (1:500, A-21206, Invitrogen). Cell nuclei were counterstained with DAPI. Images were obtained with Olympus FV1000 confocal microscope.

**RNA pull-down and MS analysis**. We performed RNA pull-down assays based on the Pierce™ Magnetic RNA-Protein Pull-Down Kit (20164, Thermo Fisher) instructions. Briefly, *LINC00842* or its antisense sequence, and 5′-end mutant *LINC00842* were transcribed in vitro using the MEGAscript® Kit (AM1333, Life Technologies) and biotinylated using the Pierce™ RNA 3′ End Desthiobiotinylation Kit (20163, Thermo Fisher). Biotinylated RNA was incubated with protein extract from PDAC cells. Streptavidin beads were then added and total proteins associated with *LINC00842* or antisense *LINC00842* were subjected to MS or Western blot analysis. In vitro transcription of *LINC00842* and its truncated fragments were produced with primers containing the T7 promoter sequence (Supplementary Table 4).

**RNA electrophoretic mobility shift assays**. Assays were performed using the LightShift Chemiluminescent RNA EMSA Kit (Thermo Fisher Scientific, Waltham, MA, USA), with biotin-labeled *LINC00842* sense strand transcribed in vitro using the MEGAscript® Kit (AM1333, Life Technologies) and biotinylated using the Pierce™ RNA 3′ End Desthiobiotinylation Kit (20163, Thermo Fisher). Briefly, 1 μl biotin-labeled RNA (4 nM final concentration) were incubated with different concentrations of PGC-1α precipitated with anti-Flag antibody from lysate of PANC-1 cells transfected with Flag-*PPARGC1A* expression vector in binding buffer (10 mM HEPES pH 7.3, 20 mM KCl, 1 mM $MgCl_2$, 1 mM dithiothreitol, 5% glycerol, and 40 U/ml RNasin) at room temperature for 30 min. The RNA–protein mixtures were separated in 5% native polyacrylamide gels at 4 °C for 1 h. The gel was transferred to a nylon transfer membrane, cross-linked to the membrane using the UVP cross-linker (120 mJ/cm$^2$ of 254 nm UV). Intensities of gel band were detected by chemiluminescence.

**Protein immunoprecipitation**. Cells or tissues were lysed in immunoprecipitation lysis buffer provided in Pierce™ Crosslink Magnetic IP/Co-IP Kit (Thermo Fisher) supplemented with 5 mM of nicotinamide, 5 μM of trichostatin A, protease, and phosphatase inhibitors for 30 min on ice. Lysate was incubated with the indicated antibodies cross-linked to Protein A/G magnetic beads for 1 h at room temperature before washing three times in immunoprecipitation lysis buffer and then eluted.

**Western blot assays**. Protein extract from PDAC cells, tissue samples, RNA pull-down, or immunoprecipitation samples were prepared using detergent-containing lysis buffer. Total protein (30 μg) was subjected to SDS-PAGE and transferred to PVDF membranes (Millipore). Membranes were then incubated overnight at 4 °C with primary antibody and visualized with a Phototope Horseradish Peroxidase Western Blot Detection kit (WBKLS0100, Millipore). Antibodies: rabbit anti-H2AFZ antibody (WB: dil. 1:1000, Cell Signaling Technology, 2718); rabbit anti-FBL antibody (WB: dil. 1:1000, Cell Signaling Technology, 2639); rabbit anti-HDAC1 antibody (WB: dil. 1:1000, Cell Signaling Technology, 34589); rabbit anti-acetyl-Lys antibody (WB: dil. 1:1000, Cell Signaling Technology, 9441); rabbit anti-YY1 antibody (WB: dil. 1:1000; 10 μg for ChIP, Cell Signaling Technology, 46395); rabbit anti-FASN antibody (WB: dil. 1:1000, Cell Signaling Technology, 3180); rabbit anti-GAPDH antibody (WB: dil. 1:2000, Cell Signaling Technology, 5174); rabbit anti-HIST1H2AG antibody (WB: dil. 1:2000, Invitrogen, PA5-24822); Rabbit anti-HIST1H1E antibody (WB: dil. 1:2000, Invitrogen, PA5-31908); rabbit anti-HIST2H2BF antibody (WB: dil. 1:2000, Invitrogen, PA5-44511); rabbit anti-SNRPE antibody (WB: dil. 1:2000, Invitrogen, PA5-96342); rabbit anti-GCN5 antibody (WB: dil. 1:1000; 5 μg for IP, Invitrogen, MA5-14884); rabbit anti-PGC-1α antibody (WB: dil. 1:1000; 5 μg for IP and RIP, Novus Biologicals, NBP1-04676); mouse anti-SIRT1 antibody (WB: dil. 1:1000; 5 μg for IP and RIP, Abcam, ab110304); mouse anti-Flag tag antibody (WB: dil. 1:2000; 5 μg for RIP, Sigma, F1804); mouse anti-β-ACTIN antibody (WB: dil. 1:20,000, Proteintech, 66009-1-Ig) (dil., dilution). Uncropped blots are provided in Supplementary Fig. 16. Image J software (1.50i) was used for immunoblots quantification.

**RNA immunoprecipitation (RIP) assay**. We performed RIP using the Magna RIP RNA-Binding Protein Immunoprecipitation kit (17-700, Millipore). Total RNA (input control) and precipitation with the isotype control (IgG) for each antibody against PGC-1α and Flag were assayed simultaneously. The co-precipitated RNAs were determined by qRT-PCR.

**Chromatin isolation by RIP (ChIRP)**. ChIRP assays were performed with the EZ-Magna ChIRP RNA Interactome Kit (17-10495, Millipore). Antisense DNA probes targeting the *LINC00842* sequence were designed using online designer at Stellaris (https://www.biosearchtech.com). We chose 24 probes uniquely spanning the entire *LINC00842* sequence, which were divided into odd and even sets according to their order. Probe sets targeting LacZ RNA were used as a negative control. All probes were 3′ biotin-labeled and the sequences are shown in Supplementary Table 4. Cells were cross-linked with 1% glutaraldehyde and quenched by glycine, followed by lysed and sonicated. Cell lysate was hybridized with odd or even sets of *LINC00842* probes or LacZ probes at 37 °C for 4 h. Magnetic streptavidin beads were then added to each hybridization reaction and rotated for 30 min. The bead–probe–RNA complex was captured with magnetic racks and 20% of the sample was used for RNA isolation while 80% for protein isolation. *LINC00842* was detected by qRT-PCR while PGC-1α was detected by western blotting.

**PGC-1α deacetylation assays**. We conducted in vitro deacetylation assay based on previously reported method[21] with some modifications. Briefly, PGC-1α was precipitated with anti-Flag antibody cross-linked to beads from lysate of 293T cells transfected with Flag-*PPARGC1A* expression vector for 24 h. After stringent washing, the eluted PGC-1α was incubated with recombinant human deacetylase SIRT1 (rhSIRT1, ab101130, Abcam) in reaction buffer containing 50 mM of Tris-HCl (pH 9.0), 50 mM of NaCl, 4 mM of $MgCl_2$, and 0.5 mM of dithiothreitol with or without 100 μM of $NAD^+$, 20 pmol of *LINC00842* sense or *LINC00842* antisense for 3 h at 30 °C. The reaction mixture was then analyzed using western blot with antibody against acetyl-Lys for the levels of acetylated PGC-1α. Flag was also blotted as a loading control marker.

**Chromatin immunoprecipitation (ChIP) assays**. ChIP assays were performed using the EZ-Magna ChIP™ A/G Chromatin Immunoprecipitation Kit (17-10086, Millipore). Briefly, PDAC cells were cross-linked with 1% formaldehyde, lysed and sonicated on ice. Pre-immunoprecipitated lysate of each sample was saved as input. Lysates were then immunoprecipitated with 5 μg of ChIP-grade antibody against YY1 or IgG as negative control. DNA was eluted, purified, and analyzed by qRT-PCR with the specific primers (Supplementary Table 4).

**Reporter gene assays**. The *LINC00842* gene promoter sequence containing YY1 binding site (−2000 bp to transcription start site) or its mutated forms was cloned into the pGL4.10 vector (Promega). The constructs were co-transfected with the pRL-SV40 Renilla vector (Promega) into PDAC cells. The luciferase activity was determined 48 h after transfection by a Dual-Luciferase Reporter Assay System (Promega) and normalized using Renilla luciferase activity.

**Statistical analysis**. We used Student *t*-test to examine the significance of the difference between two means. Fisher's exact test was used for any independence test between two categorical variables and Wilcoxon rank-sum test was used for any independence test between a continuous variable and a binary categorical variable, when there was no covariate to adjust for. Spearman's rank correlation

coefficient was used to measure the correlation between two continuous variables and $r > 0.3$ and $P < 0.05$ was considered significant. We used the log-rank test in univariate survival analyses and the Cox proportional hazards model in multivariate survival analyses. The Kaplan–Meier plot was used for presentation. Photoshop CS6 (Adobe) was used for image integration. Statistical analyses were performed using Microsoft Excel 2019, GraphPad Prism 8.0.1 (GraphPad, La Jolla, CA, USA), or the SPSS software package (version 22.0; IBM SPSS). $P < 0.05$ was considered significant for all statistical analyses and FDR was used to adjust multi comparison testing.

**Reporting summary**. Further information on research design is available in the Nature Research Reporting Summary linked to this article.

## Data availability

The accession number for the RNA-sequencing data is GSE150216. The proteomics datasets of LINC00842 sense and antisense strand pull-down assays on the web server: http://download.omicsbio.info/files/LINC00842/. Publicly available sources: RNA-sequencing data of 177 PDAC patients were retrieved from TCGA PAAD sample level 3 access (http://gdac.broadinstitute.org, version 2016_01_28; PMID: 28810144) and lncRNAs were annotated by Ensembl (PMID: 31691826); RNA array data and clinical information of PDAC patients were also downloaded from ICGC (https://dcc.icgc.org/releases/release_28/Projects/PACA-AU). Unprocessed gel blot of Figs. 4c, e, g, i, 5a–d, g–i, 6e–i, m, n, p–r, 7a and Supplementary Figs. 1c, g, 5b, 6b–d, e–g, i, 14e, f, 15b are provided in Supplementary Fig. 16. Source data are provided with this paper.

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

## Acknowledgements

This study was supported by the Program for Guangdong Introducing Innovative and Entrepreneurial Teams (2017ZT07S096 to D.L.), Natural Science Foundation of China (81772586 and 82072617 to J. Zheng, 91753142 to D.L., 81802407 to X.H.), National Young Top-notch Talent Support Program (to J. Zheng), Natural Science Foundation of Guangdong Province (2018A030313327 to X.H.), China Postdoctoral Science Foundation (2017M622880 to X.H.) and Sun Yat-sen University Intramural Funds (to D.L. and to J. Zheng).

## Author contributions

J. Zheng and D.L. conceptualized and supervised this study. X.H. and L.P. performed most experiments. M.L., R.L., G.W., L.W., Y.Z., J.S., J.D., S.D., L. Zeng, C. Wang, X.C., C. Wu, and R.C. were responsible for sample preparation, clinical data collection, and association analysis. J. Zhang, R.B., and L. Zhuang provided technique supports. M.L., L. Zeng, S. Zhang, and S. Zhu contributed to histopathological examination. Z.Z. and Y.Y. were engaged in statistical and bioinformatics analyses. J. Zheng, X.H., and D.L. prepared the manuscript. All authors reviewed the manuscript.

## Competing interests

The authors declare no competing interests.
