## [Peer Review File · Nature Communications]

REVIEWER COMMENTS

Reviewer #1 (Remarks to the Author):

Huang et al. have mined TCGA PDAC RNA-seq data and identify LINC00842 as having enrichment in PDAC. Figure 1 demonstrates clinical correlation of LINC00842 with patient outcomes and functional studies with shRNA suppression and overexpression in vitro and in vivo. Figure 2-3 then looks at transcriptional changes and metabolic changes. I do not have specific expertise on metabolism so cannot fully comment on these figures. Figure 4 shows an impressive number of pull down experiments LINC00842 binding PPARGC1A. Figure 5 shows IP for PPARGC1A with decreased acetyl-PPARGC1A in LINC-KD cell lines and relative enrichment in the LINC-OE cell line and specifically preventing SIRT1 deacetylation of PPARGC1A. Figure 6 shows high glucose leads to induction of LINC00842 again connecting metabolism with LINC00842 function. In addition, transcription factor YY1 is identified as the driver of LINC00842 to high glucose levels. Figure 7 shows a tremendous amount of data showing YY1 and FASN association with LINC00842 and acetyl-PPA levels. Finally, therapeutic ASO-LINC and Orlistat (FASN inhibitor) in animals reveal anti-tumor effects.

Questions/Comments:

- 1) The authors show survival differences in TCGA and then in their internal data from Sun Yat-sen Memorial Hospital. Does this hold up if you look at ICGC?
- 2) How is qRT-PCR used for copy number assessment (Supp Fig 1)? Is this genomic DNA or is this just RNA expression relative to something? Or is this ddPCR?
- 3) Is LINC00842 sequence conserved across different species?
- 4) Given the apparent interaction with key proteins, is there evidence of tertiary structure of the RNA with cleavage with RNAses that degrade dsRNA structures or do the authors have other indication that the interactions are with other RNA binding proteins?

Reviewer #2 (Remarks to the Author):

Huang et al. identify a long intergenic non-coding RNA (lincRNA) LINC00842 that modulates PDAC metabolism and tumor progression. The authors show that LINC00842 binds to PGC-1a (the authors should use nomenclature in the paper when referred to PPARGC1a- which should be indicated only for gene or mRNA), preventing SIRT1 deacetylation, resulting in a reprogrammed metabolic phenotype toward lipid synthesis from glucose-derived carbons in LINC00842 overexpressing PDAC cell lines. Further, they show that the transcription factor YY1 is involved in modulating LINC00842 expression in a glucose dependent manner. Treatment with a LINC00842 inhibitor and a fatty acid synthase inhibitor was found to be efficacious in orthotopic cell line and patient derived xenograft mouse models of PDAC. Although the manuscript has some interesting findings, the mechanistic basis that the authors conclude are not supported by the results. In particular, the mechanistic and functional experiments on PGC-1a acetylation/SIRT1 are poorly controlled and of difficult interpretation.

- 1- The binding of LINC00842 to endogenous PGC-1a needs further validation with mutations on LINC00842. This is important because the middle region of PGC-1a does not contain a predicted RNA binding, but the C-terminal contain the SR and RMM known to bind RNAs. It is only counterintuitive the binding as positive charges on K would likely favor negative phosphate charges on RNAs.
- 2- All the acetylation blots when endogenous or overexpressed PGC-1a and mutant alleles, or the levels should be validated keeping in mind that PGC1a full length runs around 110-120 Kd. This is important because most of the commercial antibodies are problematic.
- 3- YY1 is also known to bind RNA and PGC-1a in an m-TOR dependent manner, the dissection between the promoter activity and the binding to PGC-1a needs to be addressed and requires important controls. The induction by glucose should be re-evaluated as the concentrations used are non-physiological.
- 4- The final model that PGC-1a acetylated favors fatty acid synthesis is also problematic and needs

to be evaluated with manipulation of PGC-1a.

5- It is also very important to use a mutant allele of PGC-1a (R13 used in the literature that most of the acetylation sites are mutated from K to R. This is critical to their model.

6- In the introduction, the authors need to include critical background on PPARGC1A transcriptional activities, particularly in the context of acetylation status? It may be helpful to include a paragraph on PPARGC1A, in the context of PDAC, metabolic reprogramming, etc. It is also not clear how they build their model on acetylation-

7- The authors indicate that in LINC00842 silenced cells, the depletion of PPARGC1A reversed the effects on cell proliferation, invasion, and migration, along with restoration of metabolic phenotypes (Lines 209-214). However, the authors do not test how overexpression of LINC00842 and/or PPARGC1A including acetylation defective affected these responses. Although it is mentioned that LINC00842 overexpression did not alter mRNA levels of PPARGC1A, it may be important to determine if overexpression of PPARGC1A and accompanied overexpressed LINC00842 exacerbates the reprogrammed metabolic phenotypes in PDAC. Can the authors comment on this aspect of metabolic regulation in PDAC?

8- Were the authors able to identify a link between PDAC patient survival and PPARGC1A expression? From TCGA datasets or previously published data?

9- The authors show that FASN and LINC00842 inhibitors reduced tumor burdens in PDAC xenograft models (Lines 279-292). Can the authors comment or provide evidence that the LINC inhibitor works on the PPARGC1A acetylation axis?

Reviewer #3 (Remarks to the Author):

Summary:

In this study, Huang et al. identify human long noncoding RNA LINC00842 as novel regulator of pancreatic carcinoma development. Starting from public TCGA data on lncRNA alterations in human PDAC, LINC00842 correlated with overall PDAC survival and disease malignancy. Loss- and gain of function studies convincingly showed that LINC00842 favored cancer cell proliferation, malignancy and PDAC xenograft growth and reduced survival in cancer-bearing mice. The authors applied (targeted) metabolomics to demonstrate that LINC00842 remodels cellular metabolism and supported anaplerotic (cytoplasmic) lipid biosynthesis at the expense of TCA-mediated substrate catabolism and glucose oxidation in mitochondria. RNA-RBP (domain-specific) interaction studies elegantly defined (acetylated) PPARGC1A as interactor of LINC00842 and demonstrated that LINC00842 sterically competes with SIRT1 deacetylase for PPARGC1A repression domain binding. Glucose-dependent transcription factor YY1 was implicated in driving LINC00842 expression by performing luciferase-guided promoter analysis. Finally, the (pre)-clinical significance of their previous findings was shown using orthotopic implantation and PDX mouse models where the synergistic improvement of cancer survival was shown following fatty acid biosynthesis and / or LINC00842 inhibition.

Overall impression:

This work is of high significance to the field (and beyond) and deserves to be published in a visible scientific journal like Nature Communications. This work presents several novel aspects, for instance on the clinical significance of lncRNA alterations in cancer, and provides important mechanistic insights on how lncRNA regulate metabolic pathways and are implicated in transcriptional regulation (in this case PPARGC1A). The authors present an impressive body of evidence to support their (justified) claims and performed in vitro (and in vivo) assays with high scientific rigour. My main concerns are concerning the nature of the immortalized LINC00842-deficient cell clones, and whether potential clonality artefacts were sufficiently considered.

Major points:

L114: LINC00842-silenced cells are generated by clonal expansion of lncRNA-negative cells. What was used as control clone, and did this control clone also undergo clonal selection?

This is important to obviate a scenario where clonal selection gives rise to less malignant cell clones per se due to the selection procedure. Clonality artefacts can occur if the control clone represents the 'paternal' cell population, where the silencing clones are derived from very few (and potentially very different) sub-clones.

L131: Given that KD1 and KD2 differ in their OCR (s. Fig.2d): Which gene alterations / GO terms / are shared between KD1 and KD2 clones (vs KD Control? This is important because pVs of KEGG categories are rather low (-log₁₀(PV) 2-4). What are the global expression changes / GO terms upon LINC-OE (Fig.2c only shows selected genes)?

Minor points:

L93: More information about the expression of PDAC-associated lncRNAs should be given (mean expression (TPM), expressed in how many tissues). Also a representative browser track image of the LINC00842 locus should be shown.

L100: The number of patients should be given in the main text.

L113: Their approach to silence LINC00842 is called a 'knockdown' in Fig.S1. This term is misleading and usually referred to RNAi. Should be called transcriptional inactivation (TI) as the act of transcription is inhibited by including a poly-A stop signal through the paper.

L149: Until this point, 2xTI clones were tested, why is only one (and which) KD investigated in terms of metabolomic fluxes?

L154: If possible PDH activity should be determined to make that point clear experimentally. Same for PC activity.

L162: How is this controlled - are CTP mRNA or protein levels altered?

L171: A lncRNA marker for cellular localisation (eg MALAT for nucleus) should be included.

L181: Why should PGC1A and LINC00842 mRNA interact in the nucleus (FISH)? I assumed the colocalisation referred to PGC1A protein and LINC00842 RNA? The methods part here is very short and lacks critical detail.

L190: Description of PPARGC1A as 'PPA' is confusing in the figure.

L194: Is the main acetylation site present in the PPARGC1A repression domain? This should be described more clearly.

L198: What is the SIRT1 binding domain in PPARGC1A? Is this the same PPARGC1A Repression Domain bound by LINC00842?

L245: How does this go about mechanistically? Are Glu transporters affected by LINC00842?

Responses to The Reviewers' Comments

Reviewer #1

Huang et al. have mined TCGA PDAC RNA-seq data and identify LINC00842 as having enrichment in PDAC. Figure 1 demonstrates clinical correlation of LINC00842 with patient outcomes and functional studies with shRNA suppression and overexpression in vitro and in vivo. Figure 2-3 then looks at transcriptional changes and metabolic changes. I do not have specific expertise on metabolism so cannot fully comment on these figures. Figure 4 shows an impressive number of pull down experiments LINC00842 binding PPARGC1A. Figure 5 shows IP for PPARGC1A with decreased acetyl-PPARGC1A in LINC-KD cell lines and relative enrichment in the LINC-OE cell line and specifically preventing SIRT1 deacetylation of PPARGC1A. Figure 6 shows high glucose leads to induction of LINC00842 again connecting metabolism with LINC00842 function. In addition, transcription factor YY1 is identified as the driver of LINC00842 to high glucose levels. Figure 7 shows a tremendous amount of data showing YY1 and FASN association with LINC00842 and acetyl-PPA levels. Finally, therapeutic ASO-LINC and Orlistat (FASN inhibitor) in animals reveal anti-tumor effects.

Comment 1: The authors show survival differences in TCGA and then in their internal data from Sun Yat-sen Memorial Hospital. Does this hold up if you look at ICGC?

Response 1: Thanks for the comment, we have investigated the correlation between the LINC00842 levels in PDAC and survival time in patients in ICGC database. We are happy to report that the results are consistent with the results we found in TCGA and our patient cohort. In ICGC, patients with high LINC00842 level (\geq median) had shorter survival time than those with low LINC00842 level ($<$ median). We have added this information to revised manuscript (line 117–119; Supplementary Fig. 1b).

Comment 2: How is qRT-PCR used for copy number assessment (Supp Fig 1)? Is this genomic DNA or is this just RNA expression relative to something? Or is this ddPCR?

Response 2: We apologize for missing description of the experimental methods. This is just absolute LINC00842 RNA copy number per cell. In revised manuscript, we have detailed the detecting methods (lines 495–502) and the results are shown in Supplementary Fig. 1d.

Comment 3: Is LINC00842 sequence conserved across different species?

Response 3: We have looked at the conservation of LINC00842 among placental mammals by using PhastCons, UCSC (<http://genome.ucsc.edu/>) and the results show that LINC00842 is moderately conserved among placental mammals compared with its nearby highly conserved coding gene ANXA8L1. This information has been added to revised manuscript (line 113; Supplementary Fig. 1a).

Comment 4: Given the apparent interaction with key proteins, is there evidence of tertiary structure of the RNA with cleavage with RNAses that degrade dsRNA structures or do the authors have other indication that the interactions are with other RNA binding proteins?

Response 4: Thanks for the comment, we have performed several experiments to address this issue. Firstly, we have performed RIP assays in samples treated with RNase III or RNase A to examine the tertiary structure of LINC00842 required for PGC-1 α interaction and the result shows

that treatment with RNase A but not RNase III abolished the interaction of LINC00842 with PGC-1 α . Since ssRNA is sensitive to RNase A but resistant to dsRNA specific RNase III, this result indicates that LINC00842 may interact with PGC-1 α in a single strand manner. Secondly, we have performed Co-IP assays and revealed that the interaction between PGC-1 α and SIRT1 can be restored by RNase A but not RNase III treatment where LINC00842 is overexpressed. Together, these additional experiments have provided convincing evidence that the interaction is with single strand LINC00842. We have added these additional results to revised manuscript (lines 210–212 and Supplementary Fig. 6a; line 229–230 and Supplementary Fig. 6f).

Previous study (*Nature* 2005; 434: 113–118) has reported that acetylated PGC-1 α keeps in an inactivation status until it is deacetylated and activated by SIRT1. Acetylated PGC-1 α cannot recruit chromatin-remodelling complex (*Int J Cancer* 2019; 145: 1475–1483). In another word, acetylated PGC-1 α may not interact with other proteins. In the present study, by RNA EMSA, we found that LINC0082 interacts with PGC-1 α to form an RNA-protein complex and this additional result has been added to revised manuscript (lines 214–215; Supplementary Fig. 6b). Based on our results of RNA pull-down, RIP and ChIRP assays (Fig. 4c–e; Fig. 5e–i), which shows LINC00842 interacts with acetylated PGC-1 α , we believe that LINC00842 is likely to interact with PGC-1 α without any other RNA binding proteins.

Reviewer #2

Huang et al. identify a long intergenic non-coding RNA (lincRNA) LINC00842 that modulates PDAC metabolism and tumor progression. The authors show that LINC00842 binds to PGC-1a (the authors should use nomenclature in the paper when referred to PPARGC1a- which should be indicated only for gene or mRNA), preventing SIRT1 deacetylation, resulting in a reprogrammed metabolic phenotype toward lipid synthesis from glucose-derived carbons in LINC00842 overexpressing PDAC cell lines. Further, they show that the transcription factor YY1 is involved in modulating LINC00842 expression in a glucose dependent manner. Treatment with a LINC00842 inhibitor and a fatty acid synthase inhibitor was found to be efficacious in orthotopic cell line and patient derived xenograft mouse models of PDAC. Although the manuscript has some interesting findings, the mechanistic basis that the authors conclude are not supported by the results. In particular, the mechanistic and functional experiments on PGC-1a acetylation/SIRT1 are poorly controlled and of difficult interpretation.

Response: Thanks for the general comments. We have performed additional experiments and obtained more results to support our conclusions in revised manuscript. Per the comment, we have changed the nomenclature of PGC-1 α and PPARGC1A when refer to protein and RNA, respectively, in the revision.

Comment 1: The binding of LINC00842 to endogenous PGC-1a needs further validation with mutations on LINC00842. This is important because the middle region of PGC-1a does not contain a predicted RNA binding, but the C-terminal contain the SR and RMM known to bind RNAs. It is only counterintuitive the binding as positive charges on K would likely favor negative phosphate charges on RNAs.

Response 1: Per the suggestion, we have performed bioinformatics analysis of the interaction between LINC00842 5'-end (from nucleotide 1 to 690) and PGC-1 α using PRIdictor (<http://www.rna-society.org/raid/PRIdictor.html>) and RPISeq (<http://pridb.gdcb.iastate.edu/RPISeq/index.html>). The results showed that 6 positions in the 5'-end are likely to interact based on their Random Forest classifiers >0.5. We have then performed mutation assays and found that mutations in all these positions substantially abolish the interaction of LINC00842 with PGC-1 α as

determined by RNA pull-down assays. We have added these results in revised manuscript (line 207–210 and Supplementary Fig. 5a and 5b).

Although the R/S and RMM domains in the C-terminal of PGC-1 α may bind to RNA, our truncation assays with deletion of PGC-1 α C-terminal containing the R/S and RMM domains do not support the interaction in these regions (Fig. 4h and 4i in original manuscript). In contrary, we have observed that the Repression domain in PGC-1 α is required for the interaction (Fig. 4h and 4i). Our RIP and ChIRP assays (Fig. 5e–5i) demonstrated that LINC00842 interacts with acetylated PGC-1 α , suggesting that the interaction of LINC00842 with PGC-1 α in the Repression domain is likely due to that this domain contains acetylation sites (9 out of 13 sites).

To address whether the interaction between LINC00842 and PGC-1 α is dependent on charge, we have constructed a mutant PGC-1 α containing lysine to arginine mutation in 13 acetylation sites. Although both lysine and arginine contain positive charge, arginine residue cannot be acetylated. Immunoprecipitation assays showed that the mutations substantially abolished PGC-1 α acetylation. Subsequent RIP and RNA pull-down assays showed that PGC-1 α arginine mutants had substantially decreased interaction with LINC00842 RNA. Furthermore, RNA EMSA also showed that LINC0082 interacts with acetylated PGC-1 α to form an RNA-protein complex. These results demonstrated that the interaction of LINC00842 with PGC-1 α may not depend on the charge in amino acid residues. We have added these additional results to revised manuscript (lines 214–215, Supplementary Fig. 6b; lines 238–244, Supplementary Fig. 6g–i).

Comment 2: All the acetylation blots when endogenous or overexpressed PGC-1a and mutant alleles, or the levels should be validated keeping in mind that PGC1a full length runs around 110-120 Kd. This is important because most of the commercial antibodies are problematic.

Response 2: We have checked again and verified that the PGC-1 α and acetylated PGC-1 α blots are molecular-weight correct (Supplementary Fig. 6c). In revised manuscript, we have indicated most blots with molecular weight marker where it is possible and necessary and all uncropped immunoblot images are provided in the 'Source Data'.

Comment 3: YY1 is also known to bind RNA and PGC-1a in an m-TOR dependent manner, the dissection between the promoter activity and the binding to PGC-1a needs to be addressed and requires important controls. The induction by glucose should be re-evaluated as the concentrations used are non-physiological.

Response 3: YY1 has been reported to interact with PGC-1 α to regulate mitochondrial gene expression in an mTOR dependent manner (*Nature* 2007; 450: 736–740). To address your issues, we have performed additional assays. We have found that overexpression of YY1 in PDAC cells significantly increases the expression of LINC00842 but knockdown of PPARGC1A expression does not affect LINC00842 level. On the other hand, silencing YY1 expression significantly decreases LINC00842 level but overexpression of PPARGC1A does not alter LINC00842 level. Furthermore, reporter gene assays show that disturbing PPARGC1A expression does not alter YY1-mediated expression of reporter gene with LINC00842 promoter. These results indicate that it is YY1 that is responsible for high glucose induced LINC00842 expression in PDAC cells, which might be another regulation mechanism different from that reported previously (*Nature* 2007; 450: 736–740). We have added these additional results to revised manuscript (lines 317–323, Supplementary Fig. 13a–d).

As for glucose concentration, we used 5 mM and 25 mM in our study. We believe they are adequate. The concentration of 5 mM usually represents physiological condition and 25 mM usually used to mimic hyperglycemia condition in cell cultivation, which is often seen in literature,

for example, *Cell Metab* 2014, 19: 246–258; *Nat Chem* 2018, 10: 1103–1111; *Nat Commun* 2018, 9: 1306; *Science* 2012, 338: 1069–1072; *Mol Cell* 2011, 42: 719–730 and *Mol Cell* 2013, 49: 474–486. Importantly, *LINC00842* expression can be stimulated by both low and high glucose concentrations.

Comment 4: The final model that PGC-1 α acetylated favors fatty acid synthesis is also problematic and needs to be evaluated with manipulation of PGC-1 α .

Response 4: We apologize for the misunderstanding. Acetylated PGC-1 α is an inactive form. Previous study has demonstrated that acetylated PGC-1 α keep in an inactivation status until it is deacetylated and activated by SIRT1 (*Nature* 2005 Mar 3; 434(7029): 113–118). This is due to acetylated PGC-1 α cannot recruit chromatin-remodeling complex (*Int. J. Cancer* 2019 Sep 15; 145(6): 1475–1483). A study has reported depletion of PGC-1 α can make prostatic cancer cells from catabolic processes switch to anabolic processes, thus promote prostatic cancer progression (*Nat. Cell Biol.* 2016 Jun; 18(6): 645–656). Similar to our findings, we demonstrated overexpression of *LINC00842* can interact with acetylated PGC-1 α , making PGC-1 α maintain a high acetylation inactive status, which cause similar metabolic remodelling to depletion of PGC-1 α (see the model below).

Therefore, the metabolic switch from catabolic processes (tumor suppression) to anabolic processes (tumor progression) mediated by *LINC00842* in PDAC cells maybe caused by the amount of active PGC-1 α . In other word, *LINC00842* low expression decreases the amount of acetylated PGC-1 α , result in growing number of active PGC-1 α accumulate in PDAC cells, and cells depend on catabolic processes (tumor suppression); conversely, *LINC00842* high expression accumulates large amount of acetylated PGC-1 α , result in decreasing active PGC-1 α in PDAC cells. At this point PDAC cells switch to anabolic processes (tumor progression). As per the comment, we have also performed additional experiments to evaluate whether acetylated PGC-1 α favors fatty acid synthesis, although it has been known that PGC-1 α is involved in fatty acid synthesis (*J Hepatol*

2014, 61: 1151–1157; *Cancer Res* 2011, 71: 6888–6898; *FASEB J* 2010, 24: 1003–1014; *J Biol Chem* 2010, 285: 32793–32800). We have observed that in PDAC cells, overexpression of PGC-1 α significantly decreases but knockdown of PGC-1 α significantly increases the levels of SLC25A1 and FASN, two proteins that are important in fatty acid synthesis. Nile red staining further indicates that PGC-1 α overexpression significantly decreases but its silence significantly increases cellular lipids. These results support our notion in final model that PGC-1 α acetylated favors fatty acid synthesis. These additional data have been incorporated into our revised manuscript (lines 249–254, Supplementary Fig. 7c–f).

Comment 5: It is also very important to use a mutant allele of PGC-1 α (R13 used in the literature that most of the acetylation sites are mutated from K to R. This is critical to their model.

Response 5: We have addressed this issue by additional experiments. Please refer our Response to your Comment 1.

Comment 6: In the introduction, the authors need to include critical background on PPARGC1A transcriptional activities, particularly in the context of acetylation status? It may be helpful to include a paragraph on PPARGC1A, in the context of PDAC, metabolic reprogramming, etc. It is also not clear how they build their model on acetylation

Response 6: Per the suggestion, we have added a paragraph about PGC-1 α acetylation and its transcriptional activities in the context of PDAC in the introduction of revised manuscript (lines 74–83).

Comment 7: The authors indicate that in LINC00842 silenced cells, the depletion of PPARGC1A reversed the effects on cell proliferation, invasion, and migration, along with restoration of metabolic phenotypes (Lines 209-214). However, the authors do not test how overexpression of LINC00842 and/or PPARGC1A including acetylation defective affected these responses. Although it is mentioned that LINC00842 overexpression did not alter mRNA levels of PPARGC1A, it may be important to determine if overexpression of PPARGC1A and accompanied overexpressed LINC00842 exacerbates the reprogrammed metabolic phenotypes in PDAC. Can the authors comment on this aspect of metabolic regulation in PDAC?

Response 7: We have addressed this issue by conducting several additional experiments, which are supportive. We found that in LINC00842 overexpressing PDAC cells, overexpression of PPARGC1A substantially suppressed proliferation, migration and invasion abilities of cells. Parallely, overexpression of PPARGC1A in such cells could resume OCR, ECAR, glucose uptake and lactate production but inhibited fatty acid synthesis. We have also demonstrated that the transcriptional activity of PPARGC1A is not affected by the K to R mutation in the acetylation sites and therefore has the same effect on cell proliferation, invasion and migration as wildtype of PPARGC1A. We have added these results to revised manuscript (lines 260–264, Supplementary Fig. 9a–f).

Per the suggestion, we have also performed additional experiments to look at the effects of PPARGC1A and LINC00842 overexpression on the metabolic phenotypes in PDAC cells. The results showed that overexpression of LINC00842 significantly resumed OCR, ECAR, glucose uptake, lactate production and fatty acid synthesis inhibited by PPARGC1A. These results clearly indicate that overexpression of LINC00842 indeed exacerbates the reprogrammed metabolic phenotypes. These data together with the results reported in the original manuscript further support our conclusion that PGC-1 α participates in metabolic remodelling caused by aberrant LINC00842 expression. We have added these results to revised manuscript (lines 264–267 and Supplementary

Fig. 10a–d).

Comment 8: Were the authors able to identify a link between PDAC patient survival and PPARGC1A expression? From TCGA datasets or previously published data?

Response 8: Thanks for the comment. We have looked at the TCGA and ICGC databases and found that the PPARGC1A expression levels in tumors are not correlated with survival time in patients. The consistent results from both databases further support our conclusion that the functional effect of LINC00842 on PGC-1 α is not at the transcription level of PPARGC1A. We have added this information to revised manuscript (lines 244–248 and Supplementary Fig. 7a, b).

Comment 9: The authors show that FASN and LINC00842 inhibitors reduced tumor burdens in PDAC xenograft models (Lines 279-292). Can the authors comment or provide evidence that the LINC inhibitor works on the PPARGC1A acetylation axis?

Response 9: In the original manuscript, we have shown that treatment of ASO-LINC00842 significantly repressed LINC00842 level, which increased the PGC-1 α and SIRT1 interaction. Per the suggestion, we have also examined the PGC-1 α and SIRT1 interaction in PDXs and found that ASO-LINC00842 and FASN inhibitor substantially increase the PGC-1 α and SIRT1 interaction and decreases acetyl-PGC-1 α level. Together, these results clearly indicate that FASN and LINC00842 inhibitors works on the PGC-1 α acetylation axis. We have added these data in revised manuscript (lines 349–353 and Supplementary Fig. 14e, f).

Reviewer #3

Summary:

In this study, Huang et al. identify human long noncoding RNA LINC00842 as novel regulator of pancreatic carcinoma development. Starting from public TCGA data on lncRNA alterations in human PDAC, LINC00842 correlated with overall PDAC survival and disease malignancy. Loss- and gain of function studies convincingly showed that LINC00842 favored cancer cell proliferation, malignancy and PDAC xenograft growth and reduced survival in cancer-bearing mice. The authors applied (targeted) metabolomics to demonstrate that LINC00842 remodels cellular metabolism and supported anaplerotic (cytoplasmic) lipid biosynthesis at the expense of TCA-mediated substrate catabolism and glucose oxidation in mitochondria. RNA-RBP (domain-specific) interaction studies elegantly defined (acetylated) PPARGC1A as interactor of LINC00842 and demonstrated that LINC00842 sterically competes with SIRT1 deacetylase for PPARGC1A repression domain binding. Glucose-dependent transcription factor YY1 was implicated in driving LINC00842 expression by performing luciferase-guided promoter analysis. Finally, the (pre)-clinical significance of their previous findings was shown using orthotopic implantation and PDX mouse models where the synergistic improvement of cancer survival was shown following fatty acid biosynthesis and/or LINC00842 inhibition.

Overall impression:

This work is of high significance to the field (and beyond) and deserves to be published in a visible scientific journal like Nature Communications. This work presents several novel aspects, for instance on the clinical significance of lncRNA alterations in cancer, and provides important mechanistic insights on how lncRNA regulate metabolic pathways and are implicated in transcriptional regulation (in this case PPARGC1A). The authors present an impressive body of evidence to support their (justified) claims and performed in vitro (and in vivo) assays with high scientific rigour. My main concerns are concerning the nature of the immortalized

LINC00842-deficient cell clones, and whether potential clonality artefacts were sufficiently considered.

Response: Many thanks for the kind comments and encouragement. Per the comment, we have performed several additional experiments using LINC00842 antisense oligonucleotide (ASO) to downregulate LINC00842 expression in PDAC cells. Consistent with the previous results in CRISPR/Cas9 clones, we found that depletion of LINC00842 with ASO in PDAC cells significantly suppressed cell proliferation, migration and invasion compared with ASO control. Furthermore, downregulation of LINC00842 by ASO resulted in metabolic phenotypes similar to that in CRISPR/Cas9 clones. We have added these additional results to revised manuscript (lines 141–144, Supplementary Fig. 3a–c and lines 166–167, Supplementary Fig. 3d–g). Based on the results of functional assays shown in Fig. 6g, 6i, 6r by CRISPR/Cas9 in original manuscript and Supplementary Fig. 12c, 12d, 12i, 12h, 12j by ASO in revised manuscript, we firmly believe the results in the present study are reliable.

REVIEWER COMMENTS

Reviewer #1 (Remarks to the Author):

The authors have addressed my comments for this interesting paper.

Reviewer #2 (Remarks to the Author):

The authors have done an extensive revision and have addressed most of the concerns. However, there is an important control that needs to be performed. Endogenous PGC-1 α that is detected with antibodies in western blot and immunofluorescence should be validated. In my view, it is important that the band claimed to be PGC-1 α or in immunofluorescence should be validated in the same cells with PGC-1 α depletion using siRNA or Crispr. Some PGC-1 antibodies detect a band that is very close to the same molecular size but is non-specific.

Responses to The Reviewers' Comments

Reviewer #1

The authors have addressed my comments for this interesting paper.

Response: Thank you very much.

Reviewer #2

The authors have done an extensive revision and have addressed most of the concerns. However, there is an important control that needs to be performed. Endogenous PGC-1 α that is detected with antibodies in western blot and immunofluorescence should be validated. In my view, it is important that the band claimed to be PGC-1 α or in immunofluorescence should be validated in the same cells with PGC-1 α depletion using siRNA or Crispr. Some PGC-1 antibodies detect a band that is very close to the same molecular size but is non-specific.

Response: Per your suggestion, we have performed western blot and immunofluorescence assay by using PGC-1 α siRNAs in PANC-1 and SW1990 cell lines to verify the specificity of PGC-1 α antibody. The results show that PGC-1 α protein levels are substantially decreased after PGC-1 α siRNAs treatment compare to siControl, indicating the antibody we used in the manuscript is definitely specific to PGC-1 α . These information have been added to revised manuscript (line 655–656, line 696–697; Supplementary Fig. 15a and b).